# On the spin interface distribution for
# non-integrable variants of the two-dimensional Ising model

Rafael L. Greenblatt[*] and Eveliina Peltola[†]

### Abstract

We construct a martingale observable related to the spin interface for a class of non-integrable variants of the two-dimensional Ising model and express it in terms of Grassmann integrals. Under a conjecture about the scaling limit of this object, which is similar to some results recently obtained using constructive renormalization group methods, this would imply that the distribution of the interface at criticality has the same scaling limit as in the integrable model: Schramm-Loewner evolution SLE(3).

# Contents

---

[*]Department of Mathematics, Università di Roma "Tor Vergata", Italy. `greenblatt@mat.uniroma2.it`
[†]Department of Mathematics and Systems Analysis, Aalto University, Finland;
and Institute for Applied Mathematics, University of Bonn, Germany. `eveliina.peltola@aalto.fi`

# 1  Introduction

A number of critical discrete interface processes in two dimensions are known to converge to versions of Schramm-Loewner evolution (SLE), a universal family of conformally invariant processes satisfying a domain Markov property [Sch00]. The most common strategy for proofs of this convergence, beginning with [Smi01, LSW04] and implemented for a variety of cases in [SS05, Smi06, SS09, CDCH+14, Izy17] (among others), centers around a family of "observables" depending on the domain shape and a "marked" point on the lattice, which are martingales with respect to the natural filtration of the growing interface and admit a conformally covariant scaling limit (in a suitable sense including some local uniformity of convergence). The martingale property of the observable makes it possible to use it to analyze the Loewner driving function of the interface — and there are a number of techniques which can then be used to show that convergence of the martingale observable implies convergence of the driving function, which can then be used to prove convergence of the interface itself. There are many variants in how all of the parts of this strategy are implemented in detail, which depend in various ways on the specific properties of the model being examined, and have only been rigorously carried out for a few models, including in particular the nearest-neighbor critical Ising model [CDCH+14].

For the spin interface of the critical Ising model, the "usual" observable is one of a number of closely related objects known variously as spinors, parafermions, or discrete fermions, first introduced by Kadanoff and Ceva [KC71], subsequently identified with primary fields of conformal field theories describing the scaling limits of this and related models [NK85, CR06, CHI21], and finally applied to the proof of convergence of the interface process to SLE in the works initiated by Smirnov [Smi06, Smi10, CDCH+14] also based on earlier ideas by Aizenman, Cardy, and Kenyon, among others [Aiz96, Ken00, Ken01, RC08, IC09]. The relationship between different forms of the observable is reviewed in [CCK17]. The fact that the scaling limit of this observable is conformally invariant follows by identifying it with a holomorphic function with specific boundary conditions that characterize it uniquely, which in turn follows from discrete holomorphicity properties of the discrete observable [Mer01, CS12]. Furthermore, the discrete interface process also naturally has a version of the domain Markov property, which can be seen as an explanation for the martingale property of the observable.

Customarily (as in, e.g., [CDCH+14, KS17, Che20]), the relationship of convergence of this observable to convergence of the interface is shown by a precompactness argument based ultimately on crossing estimates, which in turn are shown to hold for the Ising model using monotonicity properties related to the Fortuin-Kasteleyn-Ginibre (FKG) inequalities. Proofs along the above lines have been formulated for various forms of the Ising model with critical pair interactions corresponding to the edges of some planar graph (with criticality defined in a way associated with a planar embedding of some specific form, which also plays a crucial role in the identification of the scaling limit, as discussed most fully in [Che20]).

## Goal of this article

We consider a version of the Ising model defined by the formal Hamiltonian

$$H^\bullet(\sigma) := -J \sum_{\{v^\bullet, w^\bullet\} \in \mathcal{E}^\bullet} \sigma_{v^\bullet} \sigma_{w^\bullet} + \lambda \sum_{X \subset \mathcal{E}^\bullet} V(X) \prod_{\{v^\bullet, w^\bullet\} \in X} \sigma_{v^\bullet} \sigma_{w^\bullet}, \qquad (1.1)$$

where the first term is the nearest-neighbor interaction with coupling constant (strength) $J > 0$ and $\lambda \in \mathbb{R}$ is a parameter controlling the strength of the added multi-spin interaction (which should be thought of as fixed but small compared to $J$), and $\mathcal{E}^\bullet$ denotes the set of nearest-neighbor pairs $\{v^\bullet, w^\bullet\}$ (edges) on the square lattice, $\sigma_{v^\bullet} \in \{\pm 1\}$ are spins, and the potential $V(X)$ on edges is translation-invariant and finite-range in the sense that it is supported on sets $X$ with uniformly bounded diameter. Given an inverse temperature $\beta > 0$, the Gibbs measure associated to the Hamiltonian $H^\bullet(\sigma)$ gives weight $\exp(-\beta H^\bullet(\sigma))$ to each spin configuration $\sigma = (\sigma_{v^\bullet})$.

At least for small $|\lambda|$, it is expected that these models belong to the same universality class as the $\lambda = 0$ case (which is simply the standard Ising model on the square lattice), meaning that they exhibit an order-disorder phase transition at a unique critical temperature $1/\beta_c \in (0, \infty)$, and several properties of the critical point are the same. It has been shown [GGM12, AGG21, AGG23, CGG24] that some correlation functions indeed have the same asymptotic behavior up to rescalings, and on physical grounds it seems reasonable to expect that the scaling limit of the spin interfaces is also the same. Any proof of this will, however, have a number of differences from the standard case: generically, the model defined by Equation (1.1) lacks discrete holomorphic observables — and even the FKG inequality used to obtain crossing estimates holds only in some special cases.

In what follows, we generalize the observable defined in [CS12, Izy17] for the spin (or Peierls) interface of the Ising model on a square lattice to these non-integrable models[1]. The generalization is still a martingale with respect to the (new) discrete interface process: as we emphasize, the martingale property is a result of the way that both the interface and the observable are defined as functions of random low-temperature contour configurations, without any dependence on the distribution of these contours. We show in Theorem 4.4 that the observable can be represented in terms of correlation functions of an interacting lattice fermionic quantum field theory, that is, as a Grassmann integral with a certain form (analogously to the well-known integrable case discussed in detail in [CCK17] and references therein). Importantly, the corresponding Grassmann observable is local (like the energy observable studied in [GGM12, AGG23], or the spin on the boundary [Cav20, CGG24], but unlike the generic spin). Therefore, although we will not attempt to make any rigorous conclusions concerning the scaling limit, there are reasons to believe that a scaling limit could be eventually obtained by constructive renormalization group methods. Such

---

[1]We shall only consider the spin interface; e.g., we do not consider the FK (or random-cluster) interfaces, which for the standard Ising model converge to SLE(16/3) [CDCH+14, BPW21, Izy22, FPW24]. In fact, it is not even clear whether there is a sensible generalization of this interface to the class of models we are considering: it is defined in terms of random clusters, which are not measurable in terms of the spin configuration, but require the introduction of auxiliary random variables. This can be done for Hamiltonians consisting only of pair interactions with a fixed sign, but even then if the interacting pairs do not form a planar graph, it is not clear how to identify the boundary of such a cluster with a curve in the plane. For Hamiltonians with terms involving more that two spins, the situation is even more complicated.

a program, however, suffers from at least one significant technical obstacle concerning extending the existing versions of these methods from the case of a straight boundary considered in [Cav20, AGG23, CGG24] to very irregular domains associated with the interface process. It is in any case to be expected that the observable is simply rescaled (along with a change in the critical temperature, see Conjecture 4.5), leading to exactly the same scaling limit for the interface as in the integrable case. This should be distinguished from, e.g., interacting dimer models [GMT17], where at least some observables change more qualitatively but are still expected to have conformally invariant scaling limits.

It is important to note that convergence of the observable might, a priori, not be sufficient to guarantee convergence of the interfaces in a strong sense in a space of curves. As already mentioned, in existing proofs for the Ising model this implication is obtained using crossing estimates derived in [KS17], building on [AB99], but which are ultimately based on the FKG inequality, which does not generally hold in the non-integrable case that we consider in the present work. On the other hand, by quite general arguments originating in [LSW04], convergence of a martingale observable (in a suitable sense) implies that the Loewner driving process of the interface converges to a suitable limit. This is weaker than convergence of the interface process itself — however, under a reversibility condition a result of Sheffield and Sun [SS12] shows that it actually implies convergence of the interface. The relevant reversibility property does not hold for all interface processes, but it does hold for our construction (this depends on fixing some details of the process in Definition 2.2, which are unimportant in other contexts, in a specific way).

In light of this, it seems quite likely that at the appropriate critical temperature, the spin interface of the model defined by the Hamiltonian (1.1) converges to SLE(3), as in the familiar (planar) case with $\lambda = 0$. One feature of this conjecture, which may be surprising, is that the interface process of the discrete model does not have all of the Markov properties as in the cases which are more commonly considered. For example, in the standard Ising model on a planar graph, conditioned on the presence of a certain Peierls interface dividing the system into two parts, everything else about these two parts is (conditionally) independent. This is not the case for (1.1) in general, since the Hamiltonian can include interaction terms crossing the interface. As a result, the discrete interface process does not have the domain Markov property — which is one of the defining features of SLE. Nevertheless, what we are arguing in the present work is that the behavior of the observables conjectured in Section 4.2 still makes it possible to conclude that the full domain Markov property is restored as an *emergent property* of the scaling limit, as is conformal invariance (which also does not have an exact discrete analogue in all of the models under consideration). Related to this, when defining boundary conditions, it is necessary to specify the spin configuration at some distance from the boundary — in particular, it takes more than labelling two points on the boundary to specify "Dobrushin" type boundary conditions (hence the introduction of $P_o$ in Section 4). Part of Conjecture 4.5, which we present in Section 4.2, is that this choice plays no role in the scaling limit.

Let us also note that the present work is limited to a square lattice, whereas for the planar nearest-neighbor Ising model ($\lambda = 0$) the spin interface is known to converge to SLE(3) for a very general class of graphs [CS12, Che20]. For non-integrable interactions, the situation is likely to be significantly more complex for the following reason. The

interaction has an effect analogous to changing the coupling of the planar model (and changing the critical temperature), a priori changing each edge in a way depending on the structure the Hamiltonian. For the interface on the square lattice (and a few other regular graphs) it is possible to guarantee that this effect of the interaction is uniform (apart from boundary effects) by stipulating that it respects the symmetries of $\mathbb{Z}^2$ sufficiently well. In many other cases, the simplest plausible scenario is that the long-distance behavior of the system resembles that of an Ising model with inhomogeneous couplings, which includes cases where the critical behavior of the model is changed in dramatic ways. For instance, disordered models which are translation-invariant in one direction apparently have a critical point only in a very different sense than the familiar case [MW73, Fis95] (see also [CGG19, GG22] for recent progress towards a rigorous proof of this statement), and hence should not exhibit conformal invariance.

## Outline of the rest of this article

In Section 2, we review the definition of the Ising model with Dobrushin boundary conditions, the low-temperature contour representation, and the spin (or Peierls) interface. We do this partly to make the present work as self-contained as possible, but more so to introduce many definitions used in the rest of this article (some of which involve conventions which vary in the literature), as well as to emphasize some features of these objects which are particularly relevant to the present work. In particular, in Section 2.3 we introduce an interface exploration process (Definition 2.2) and the associated martingale observable (Proposition 2.4), which is a variant of the one used to prove the conformal invariance of the critical Ising model in the scaling limit in Smirnov's seminal works [Smi06]. We stress that in the construction we present here (based in particular on [Izy17]), the exploration process and the martingale observable are defined as functions of contour configurations in a way that guarantees the martingale property without depending on the details of the probability distribution of the contours, which will make it easy to generalize to the non-integrable systems which are the main subject of this article. In Section 3, we review (one version of) the representation of the planar Ising model in terms of Grassmann variables and give an expression for the martingale observable in this framework, i.e., as a Berezin integral. In particular, we do this using the relationship to the low-temperature contour representation in a way which prepares the later generalization.

Then, in Section 4 we arrive at the modified (non-integrable) model, which we examine first in terms of low-temperature contours and then use them to derive a Grassmann representation, including an expression for a martingale observable defined using the construction introduced in the previous sections (Equation (4.6) and Theorem 4.4). This expression has the form of a correlation function with respect to a modified (non-Gaussian) Berezin measure, similar to those for other correlation functions studied via constructive renormalization group techniques in [GGM12, Cav20, AGG21, AGG23, CGG24]. The representation obtained here is not quite the same as the one in the publications cited above, essentially because that representation is based on high-temperature contours rather than the low-temperature contours used here. As well as being more naturally connected to the spin interface and fixed or Dobrushin boundary conditions (but therefore more awkward for open boundary conditions as studied in [AGG21, AGG23, CGG24]), the current

representation is somewhat easier to derive, and has all of the relevant features for the application of constructive renormalization group methods (compare Proposition 4.2 with [GGM12, Proposition 1] or [AGG21, Proposition 3.1]). Unfortunately, so far these methods have only been implemented for toroidal and cylindrical domains, so we are not able to proceed any further in a rigorous fashion at this stage. The results that have been obtained so far do however lead to a natural conjecture about these correlation functions, which we formulate as Conjecture 4.5, and which would imply their convergence in a locally uniform fashion to the same scaling limit as in the well-studied planar case.

Let us finally remark that in this work, we do not attempt to derive or even introduce discrete Cauchy-Riemann equations [CS12, Che20], since they cannot be expected to hold in general for the models we are considering, and (unlike other subjects we review in more detail) their only role is in characterizing the scaling limit in the $\lambda = 0$ case. In fact, even in this case, the observables we shall introduce explicitly will not, strictly speaking, satisfy such equations in any familiar form without some modification (e.g., linear combination) to produce related complex-valued objects, which we avoid spelling out in detail.

### Acknowledgments

The work of R.L.G. was supported by the European Research Council (ERC) under the European Union's Horizon 2020 research and innovation programme (ERC StG MaMBoQ, grant agreement No. 802901 for R.L.G.) and by the MIUR Excellence Department Project MatMod@TOV awarded to the Department of Mathematics, University of Rome Tor Vergata, CUP E83C18000100006.

While carrying out this project, E.P. has been supported by the Deutsche Forschungsgemeinschaft (DFG, German Research Foundation) under Germany's Excellence Strategy EXC-2047/1-390685813, the DFG collaborative research centre "The mathematics of emerging effects" CRC-1060/211504053, as well as by the Academy of Finland Centre of Excellence Programme grant number 346315 "Finnish centre of excellence in Randomness and STructures (FiRST)", the Academy of Finland grant number 340461 "Conformal invariance in planar random geometry", and the European Research Council (ERC) under the European Union's Horizon 2020 research and innovation programme (101042460): ERC Starting grant "Interplay of structures in conformal and universal random geometry" (ISCoURaGe).

## 2    Preliminaries on the Ising model

In this section we introduce the low-temperature contour representation, Dobrushin boundary conditions, and the spin interface and the associated exploration process and martingale observable. While all of this content is very well known, we have chosen to present it in some detail for two reasons beyond a desire to make this article as self-contained as possible. First, we use this as an opportunity to clearly formulate some objects which are the subject of varying conventions in the literature. Second, we will underline certain features of these objects which are important for the generalizations we introduce later.

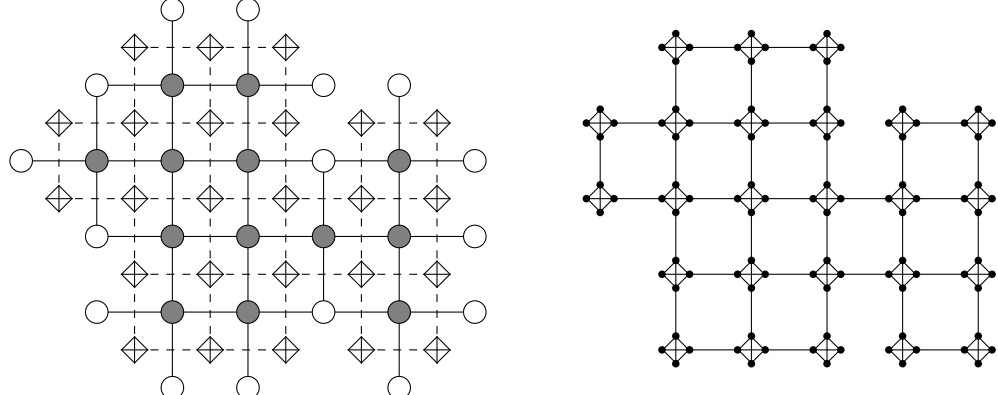

Figure 2.1: An example of the graphs being used. On the left, the points of $\Omega \subset \mathcal{V}^{\bullet}_{\Omega}$ (interior primal vertices), where the variable spins will be located, are shown as "●", the boundary vertices in $\partial\mathcal{V}^{\bullet}_{\Omega}$ (having fixed spins) as "○", and edges in $\mathcal{E}^{\bullet}_{\Omega}$ as solid lines. The faces $\mathcal{F}_{\Omega}$ (identified with dual vertices) are drawn as "◇", and the associated dual edges $\mathcal{E}_{\Omega}$ as dashed lines. On the right, the corresponding cluster graph $\mathscr{G}^{c}_{\Omega} = (\mathscr{V}^{c}_{\Omega}, \mathscr{E}^{c}_{\Omega})$.

We begin in Section 2.1 by defining the geometrical objects underlying the other definitions. Then in Section 2.2, we define the nearest-neighbor spin-Ising model, its partition functions, and the boundary conditions of interest. The low-temperature expansion of the domain walls between the variable spins gives rise to a contour representation of the Ising model, enabling us to focus on graphical quantities instead of the spins. We hence turn to a model of contour configurations (that can be defined more generally than the spin model) and define various objects related to the interface in terms of such a model in Section 2.3.

## 2.1 Primal, dual, and cluster graphs and contour configurations

We consider spin models on subsets of the two-dimensional square lattice $\mathbb{Z}^2 = (\mathcal{V}^{\bullet}, \mathcal{E}^{\bullet})$, whose vertices we identify with elements of $\mathcal{V}^{\bullet} := \mathbb{Z} + \mathrm{i}\,\mathbb{Z}$ (the set of complex numbers with integer real and imaginary parts) and edges comprise the set $\mathcal{E}^{\bullet}$ of unordered pairs $\{v^{\bullet}, w^{\bullet}\} \subset \mathbb{Z} + \mathrm{i}\,\mathbb{Z}$ with $|v^{\bullet} - w^{\bullet}| = 1$. The following definitions are illustrated in Figure 2.1.

Let $\Omega \subset \mathcal{V}^{\bullet}$ be a finite subset of the vertices and let $\mathcal{E}^{\bullet}_{\Omega} \subset \mathcal{E}^{\bullet}$ be the set of edges with at least one element in $\Omega$. Let $\mathcal{V}^{\bullet}_{\Omega} \supset \Omega$ be the set of vertices in $\mathcal{V}^{\bullet}$ which are at distance one or less from $\Omega$, and set $\partial\mathcal{V}^{\bullet}_{\Omega} := \mathcal{V}^{\bullet}_{\Omega} \setminus \Omega$. We call $\mathcal{G}^{\bullet}_{\Omega} = (\mathcal{V}^{\bullet}_{\Omega}, \mathcal{E}^{\bullet}_{\Omega})$ the *primal graph*.

Next, let $\mathcal{F}$ be the set of faces of the lattice $\mathbb{Z}^2$, which are unit squares, naturally identified with points in $(\mathbb{Z} + \frac{1}{2}) + \mathrm{i}\,(\mathbb{Z} + \frac{1}{2})$ (vertices of the dual lattice). Let $\mathcal{E}$ be the set of unordered pairs of adjacent faces in $\mathcal{F}$, i.e., dual edges. Note that each edge $e^{\bullet} \in \mathcal{E}^{\bullet}$ crosses exactly one element of $\mathcal{E}$, which we denote $e^2$; we will frequently use this one-to-one correspondence $e^{\bullet} \leftrightarrow e$. We say that $e \in \mathcal{E}$ is *incident to* a face $f \in \mathcal{F}$, and write $e \sim f$, if the corresponding dual vertex at $f$ is one of the endpoints of the dual edge $e$.

Let $\mathcal{E}_{\Omega}$ be the set of dual edges $e$ associated with edges $e^{\bullet} \in \mathcal{E}^{\bullet}_{\Omega}$, and let $\mathcal{F}_{\Omega}$ be the set of the faces incident to the dual edges in $\mathcal{E}_{\Omega}$. We call $\mathcal{G}_{\Omega} = (\mathcal{F}_{\Omega}, \mathcal{E}_{\Omega})$ the *dual graph*. We consider contour configurations on the dual graph.

---

[2]Note also that faces of the dual lattice naturally correspond to (primal) vertices $\mathcal{V}^{\bullet}$.

We will make extensive use of the representation of the Ising model in terms of Grassmann variables indexed by "half-edges", which are pairs $h = (f(h), e(h)) \in \mathcal{F} \times \mathcal{E}$. Let $\mathscr{V}_\Omega^c$ be the set of half-edges with $f(h) \in \mathcal{F}_\Omega$ (but not necessarily $e(h) \in \mathcal{E}_\Omega$). Let $\mathscr{E}_\Omega^{\mathrm{short}}$ (so-called "short edges") be the set of unordered pairs $\{h_1, h_2\} \subset \mathscr{V}_\Omega^c$ with $f(h_1) = f(h_2)$ and $e(h_1) \neq e(h_2)$, and let $\mathscr{E}_\Omega^{\mathrm{long}}$ be the set of pairs with $e(h_1) = e(h_2) \in \mathcal{E}_\Omega$ and $f(h_1) \neq f(h_2)$. The graph $\mathscr{G}_\Omega^c = (\mathscr{V}_\Omega^c, \mathscr{E}_\Omega^c)$, where $\mathscr{E}_\Omega^c := \mathscr{E}_\Omega^{\mathrm{short}} \cup \mathscr{E}_\Omega^{\mathrm{long}}$, is a version of the *cluster graph* or *terminal graph* introduced by Kasteleyn [Kas63] and illustrated in Figure 2.1. Compared with [CCK17], for example, our definition has some apparently redundant elements, which we find useful for introducing boundary conditions and for identifying elements of the graphs associated with different domains $\Omega$. Note that this graph, unlike the others introduced so far, is necessarily nonplanar. The set $\mathscr{E}_\Omega^{\mathrm{long}}$ of long edges is naturally in one-to-one correspondence with $\mathcal{E}_\Omega$, and for each face $f \in \mathcal{F}_\Omega$ there are exactly four elements of $\mathscr{E}_\Omega^{\mathrm{short}}$ which form the edges of a complete 4-graph, which we denote by $\bigoplus_f$.

The cluster graph was introduced by Kasteleyn to relate the Ising model to a dimer model in a way that is closely related to the representation we will use in this article. A *perfect matching* (a configuration of a dimer model) on $\mathscr{G}_\Omega^c$ is a collection $\{e_1^c, e_2^c, \ldots, e_n^c\} \subset \mathscr{E}_\Omega^c$ of $n = \frac{1}{2}|\mathscr{V}_\Omega^c|$ edges in which every vertex of $\mathscr{G}_\Omega^c$ appears exactly once. Let $\mathcal{P}_\Omega$ be the set of even-degree subgraphs of the dual graph, i.e., *contour configurations* $P \subset \mathcal{E}_\Omega$ on dual edges for which the *degree* $\mathrm{d}_f(P) := |\{e \in P \mid e \sim f\}|$ is even for all $f \in \mathcal{F}_\Omega$. On the cluster graph $\mathscr{G}_\Omega^c$, any perfect matching always includes an even number of long edges incident to each face $f \in \mathcal{F}_\Omega$. Hence, it can be put into (nonunique) correspondence with a contour configuration $P \in \mathcal{P}_\Omega$, by completing the perfect matching at the short edges (note that there can be several ways to match the short edges that results in the same contour configuration). On the other hand, as long as one spin on the boundary is fixed spin configurations $\sigma$ on $\Omega$ correspond bijectively to contour configurations $P \in \mathcal{P}_\Omega$ (see Section 2.2). This gives a correspondence of spin configurations on $\Omega$ and perfect matchings (dimer covers) on $\mathscr{G}_\Omega^c$, which makes it possible to express many quantities of interest in terms of Pfaffians of antisymmetric matrices related to the adjacency matrix of $\mathscr{G}_\Omega^c$ [Kas63].

## 2.2 Ising model and its low-temperature contour representation

Standard references for the Ising model include the books [MW73, Bax07]. A *spin configuration* on $\Omega$ is an assignment $\sigma \in \{\pm 1\}^{\mathcal{V}_\Omega^\bullet}$ of variable spins $\sigma_{v^\bullet}$ equaling $-1$ or $+1$ at each vertex $v^\bullet \in \mathcal{V}_\Omega^\bullet$. The *Ising model* is a Boltzmann distribution on the set of all spin configurations. Given coupling constants $\underline{J} = (J_{e^\bullet})_{e^\bullet \in \mathcal{E}_\Omega^\bullet} \in [0, \infty)^{\mathcal{E}_\Omega^\bullet}$, the (ferromagnetic, nearest-neighbor) Ising Hamiltonian with "+" boundary conditions is[3]

$$H_{\Omega, \underline{J}}^+(\sigma) = 2 \sum_{e^\bullet \in \mathcal{E}_\Omega^\bullet} J_{e^\bullet} \, \epsilon_{e^\bullet}(\sigma), \tag{2.1}$$

where $\epsilon_{e^\bullet}(\sigma) = \frac{1}{2}(1 - \sigma_{v^\bullet}\sigma_{w^\bullet})$ for edges $e^\bullet = \{v^\bullet, w^\bullet\}$, with $\sigma_{v^\bullet} = 1$ for all $v^\bullet \in \partial\mathcal{V}_\Omega^\bullet$. Let $\beta \in (0, \infty)$ be the inverse temperature and $\underline{x} = (x_{e^\bullet})_{e^\bullet \in \mathcal{E}_\Omega^\bullet}$ the collection of *edge weights*

$$x_e = \exp(-2\beta J_{e^\bullet}) \tag{2.2}$$

---

[3]This differs from the most commonly used Ising spin Hamiltonian by a constant term $-\sum_{e^\bullet \in \mathcal{E}_\Omega^\bullet} J_{e^\bullet}$.

where $e$ is the dual edge crossing $e^*$. As $\epsilon_{e^*} = \epsilon_{e^*}(\sigma) \in \{0, 1\}$ for all $\sigma$, we can write

$$\exp\left(-\beta H^+_{\Omega,\underline{J}}(\sigma)\right) = \prod_{e^* \in \mathcal{E}^*_\Omega} x_e^{\epsilon_{e^*}(\sigma)} = \prod_{\substack{e^* \in \mathcal{E}^*_\Omega \\ \epsilon_{e^*}(\sigma)=1}} x_e = \prod_{e \in P(\sigma)} x_e, \qquad (2.3)$$

using the one-to-one correspondence $e^* \leftrightarrow e$ and writing $P(\sigma) := \{e \in \mathcal{E}_\Omega \mid \epsilon_{e^*}(\sigma) = 1\}$. The range of $P$ is exactly the $\mathcal{P}_\Omega \subset 2^{\mathcal{E}_\Omega}$ which are *even* in the sense that $\mathrm{d}_f(P)$ is even for all $f \in \mathcal{F}_\Omega$. In fact each element of $\mathcal{P}_\Omega$ is associated with a unique spin configuration, so the Ising model is equivalent to one on this state space, known as the *low-temperature contour representation*. We have the partition function

$$Z^+_{\Omega,\underline{x}} := \sum_{\sigma \in \{\pm 1\}^\Omega} e^{-\beta H^+_{\Omega,\underline{J}}(\sigma)} = \sum_{P \in \mathcal{P}_\Omega} \prod_{e \in P} x_e. \qquad (2.4)$$

and the distribution of the random variable $P(e)$ is the measure

$$\mathbb{P}^+_{\Omega,\underline{x}}[P] := \frac{1}{Z^+_{\Omega,\underline{x}}} \prod_{e \in P} x_e, \qquad P \in \mathcal{P}_\Omega,$$

which we consider as being parameterized by the edge weights $\underline{x}$.

This can be generalized to Dobrushin boundary conditions as follows. For fixed $\xi = (h_{\mathrm{in}}, h_{\mathrm{out}}) \in \mathscr{V}^c_\Omega \times \mathscr{V}^c_\Omega$ with $h_{\mathrm{in}} \neq h_{\mathrm{out}}$, let $\mathcal{P}^\xi_\Omega$ be the set of $P \subset \mathcal{E}_\Omega \setminus \{e(h_{\mathrm{in}}), e(h_{\mathrm{out}})\}$ such that $P$ is even with the two indicated half-edges counted in the degree of their face, i.e.,

$$\mathrm{d}_f(P) + \mathbb{1}_{f(h_{\mathrm{in}})}(f) + \mathbb{1}_{f(h_{\mathrm{out}})} = |\{e \in P \mid e \sim f\}| + \mathbb{1}_{f(h_{\mathrm{in}})}(f) + \mathbb{1}_{f(h_{\mathrm{out}})} \in 2\mathbb{Z}, \qquad (2.5)$$
$$\text{for all } f \in \mathcal{F}_\Omega.$$

See Figure 2.2 for an example. We also write $\mathcal{P}^\emptyset_\Omega := \mathcal{P}_\Omega$ for the earlier even-degree contours.

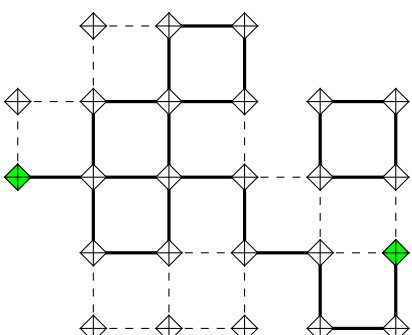

Figure 2.2: An example of a contour configuration $P \in \mathcal{P}^\xi_\Omega$ with the same $\Omega$ and $\xi$ as in the preceding figures. Here, only the marked faces $f(h_{\mathrm{in}})$ and $f(h_{\mathrm{out}})$ are highlighted as "⬙". While the marked dual edges $e(h_{\mathrm{in}})$ and $e(h_{\mathrm{out}})$ do not play any role in $\mathcal{P}^\xi_\Omega$ when they are on the boundary, they are important for the relation with Grassmann integrals (see Section 3) and for the generalized boundary conditions used to define martingale observables in Section 2.3.

We say that a pair $\xi = (f(h_{\mathrm{in}}), e(h_{\mathrm{in}})) \in \mathscr{V}^c_\Omega \times \mathscr{V}^c_\Omega$ is an *admissible boundary condition* if there exist $\Omega_o \supset \Omega$ and $P_o \subset \mathcal{E}_{\Omega_o \setminus \Omega}$ with $|\Omega_o| < \infty$ such that $P \cup P_o \in \mathcal{P}_{\Omega_o}$ for all

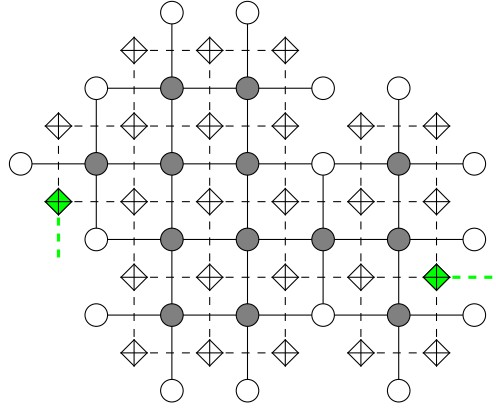 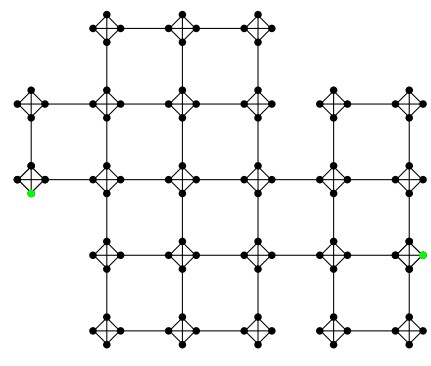

(a) An example of an admissible boundary condition $\xi = (h_{\text{in}}, h_{\text{out}})$ for the same graph as above, with the faces $f(h_{\text{in}})$ and $f(h_{\text{out}})$ shown as "⬦" and the external dual edges $e(h_{\text{in}})$ and $e(h_{\text{out}})$ in green.

(b) The corresponding cluster graph, with the half-edges $h_{\text{in}}, h_{\text{out}}$ in green.

Figure 2.3: Admissible boundary conditions.

$P \in \mathcal{P}_\Omega^\xi$. In particular this requires that $h_{\text{in}}, h_{\text{out}}$ lie on the boundary of $\Omega$ as shown in Figure 2.3. Then, $P \cup P_o$ is associated with a unique spin configuration on $\Omega_o$ and in particular, there are two components of the boundary $\partial \mathcal{V}_\Omega^\bullet$ depending only on $\xi$ which are always assigned opposite values, depending only on $P_o$, as shown in Figure 2.4, what are known as *Dobrushin boundary conditions*. In terms of the contours, the associated partition function is then given by

$$Z_{\Omega,\underline{x}}^\xi = \sum_{P \in \mathcal{P}_\Omega^\xi} \prod_{e \in P} x_e, \qquad \xi = (h_{\text{in}}, h_{\text{out}}), \tag{2.6}$$

and the associated probability measure is

$$\mathbb{P}_{\Omega,\underline{x}}^\xi[P] := \frac{1}{Z_{\Omega,\underline{x}}^\xi} \prod_{e \in P} x_e, \qquad P \in \mathcal{P}_\Omega^\xi. \tag{2.7}$$

Note that these objects do not actually depend on $\Omega_0$ and $P_o$ except via $\xi$. In fact, the definitions make perfect sense even when $\xi$ is not admissible (although the relationship to the Gibbs measure of the Ising model breaks down) and we use this generalization in an important way in the following section.

**Remark 2.1.** All of this can easily be generalized to $\xi = (h_1, \ldots, h_{2N})$ with $N \geq 2$, corresponding to alternating boundary conditions as in [Izy17], and with slightly more attention this is true for what follows as well (but we will mostly stick to the case with two marked points to avoid notational complications).

## 2.3 Exploration interface and martingale observable

The following discussion makes no direct connection to the spin Ising model; it is valid in general for any probability measure on the contour space $\mathcal{P}_\Omega^\xi$ — including the representation

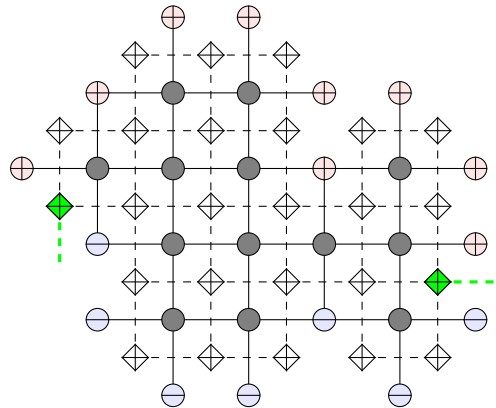

Figure 2.4: An example of a choice of the faces $f(h_{\text{in}})$ and $f(h_{\text{out}})$ (shown as "⬥") and the external dual edges $e(h_{\text{in}})$ and $e(h_{\text{out}})$ specifying a choice of Dobrushin boundary conditions. The boundary is divided into two components, drawn as "⊕" and "⊖".

of the more general model introduced in Section 4, which does not have the same form as Equation (2.7) (see in particular Theorem 4.4).

For $\xi = (h_{\text{in}}, h_{\text{out}})$ we can define the *Peierls interface*, or *exploration path* (with Dobrushin boundary conditions) as a path from a given boundary vertex $h_{\text{in}} = (f(h_{\text{in}}), e(h_{\text{in}}))$ to another vertex $h_{\text{out}} = (f(h_{\text{out}}), e(h_{\text{out}}))$ consisting of edges which are included in $P$, with no edge traversed more than once. In terms of the spins, when $\xi$ is an admissible boundary condition, one can think of the interface as separating spins of different values (however we continue to make definitions in the general case not admitting such an interpretation). For $P \in \mathcal{P}_\Omega^\xi$ with $\xi = (h_{\text{in}}, h_{\text{out}})$, the parity constraint guarantees that such a path exists, but not that it would be unique (due to an ambiguity for faces with degree $\mathrm{d}_f(P) = 4$). We shall fix a unique choice by the following procedure, illustrated in Figure 2.5.

**Definition 2.2.** We iteratively construct sequences $(\gamma_n)_{n \in \mathbb{Z}_{\geq 0}} = (\gamma_n(P))_{n \in \mathbb{Z}_{\geq 0}}$ of dual edges and $(\vec{\gamma}_n)_{n \in \mathbb{Z}_{\geq 0}} = (\vec{\gamma}_n(P))_{n \in \mathbb{Z}_{\geq 0}}$ of half-edges, starting at the initial edge $\gamma_0 := e(h_{\text{in}})$ and half-edge $\vec{\gamma}_0 := h_{\text{in}}$, and iterating as follows (see Figure 2.5).

1. If $\gamma_n = e(h_{\text{out}})$, then stop.
2. If only one dual edge $e \in P \cup \{e(h_{\text{out}})\}$ which does not already appear in $\{\gamma_0, \ldots, \gamma_n\}$ is incident to $f(\vec{\gamma}_n)$, then set $\gamma_{n+1} := e$ and set $\vec{\gamma}_{n+1} := (f', e)$, where $f'$ is the other face incident to $e$.
3. If neither of the above holds, then the parity constraint requires that all of the other edges incident to $f(\vec{\gamma}_n)$ belong to $P \cup \{e(h_{\text{out}})\}$. In this case, set $\gamma_{n+1}$ to be the edge such that $\{\gamma_n, \gamma_{n+1}\} = \{f(\vec{\gamma}_n) \pm 1/2, f(\vec{\gamma}_n) \pm \mathrm{i}/2\}$ and set $\vec{\gamma}_{n+1}$, where $f'$ is the other face incident to $\gamma_{n+1}$. (This corresponds to resolving the ambiguity by pairing edges in the interface in the North-East and South-West directions[4], and we refer to it later as the NE/SW rule and the preferred pairs of edges as NE/SW pairs).

---

[4]This is not the only way to define an interface which gives the same scaling limit. However some choices, for example the construction used in [CS12, HK13], are not reversible.

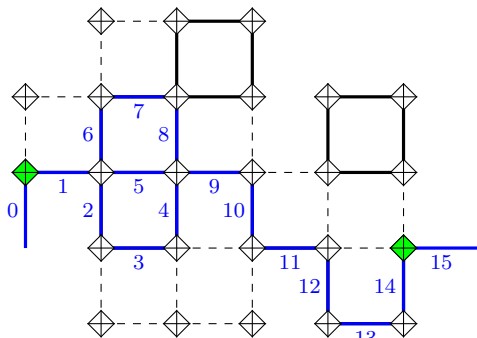

Figure 2.5: The edges forming the Peierls interface $\gamma$ associated with the contour configuration in Figure 2.2 and a compatible choice of dual edges $e(h_{\text{in}}), e(h_{\text{out}})$, shown in blue and numbered according to the construction in Definition 2.2, with ambiguities resolved according to the North-East/South-West rule. Observe that the numbering is reversed if the roles of $f(h_{\text{in}})$ and $f(h_{\text{out}})$ are reversed. In this example, we have $n_{\text{out}} = 15$.

Letting $n_{\text{out}}$ be the highest value of $n$, that is, the time when the exploration path terminates with $\gamma_{n_{\text{out}}} = e(h_{\text{out}})$, the sequence $\gamma_{[0,n]} = (\gamma_0, \gamma_1, \ldots, \gamma_{n \wedge n_{\text{out}}})$ is a Markov chain and $n_{\text{out}}$ is an associated stopping time (for the natural filtration generated by $\gamma$). Note that the interface is manifestly reversible: exchanging $h_{\text{in}}$ and $h_{\text{out}}$ gives two different Markov chains, which however are related by time-reversal with respect to $n_{\text{out}}$ as in [SS12]. In particular, the terminal time $n_{\text{out}}$ and the path $\{\gamma_1, \ldots, \gamma_{n_{\text{out}}}\}$ are unchanged as random variables. The state space of the Markov chain is the collection of all sequences $(\gamma_0, \gamma_1, \ldots, \gamma_n)$ of dual edges of various lengths $n \in \mathbb{Z}_{\geq 0}$. As the graph is assumed to be bounded, the length is bounded from above (so the state space is finite but possibly large).

Denote $2^{\mathcal{E}_\Omega} = \{X \mid X \subset \mathcal{E}_\Omega\}$. Given a family of complex-valued functions $W^\xi \colon \mathcal{P}_\Omega^\xi \to \mathbb{C}$ parameterized by $\xi \in \mathscr{V}_\Omega^c \times \mathscr{V}_\Omega^c$, for each $n \in \mathbb{Z}_{\geq 0}$, define a family of functions

$$F_W(\gamma_{[0,n]}; \cdot) \colon \mathscr{V}_\Omega^c \to \mathbb{C},$$
$$F_W(\gamma_{[0,n]}; h) := \sum_{P \in \mathcal{P}_\Omega^{h_{\text{in}},h}} \mathbb{1}_{C_{\gamma,n}}(P) \, W^{h_{\text{in}},h}(P), \qquad h = (f, e) \in \mathscr{V}_\Omega^c, \qquad (2.8)$$

with $h_{\text{in}} = \vec{\gamma}_0$, where $C_{\gamma,n}$ is the event that $\gamma_{[0,n]}$ is part of the Peierls interface, i.e.,

$$C_{\gamma,n} := \{P \subset \mathcal{E}_\Omega \mid \gamma_{[0,n]} = \gamma_{[0,n]}(P)\}.$$

A contour configuration $P \in C_{\gamma,n}$ necessarily contains $\{\gamma_1, \ldots, \gamma_n\}$, but also excludes some adjacent dual edges which would have formed part of the interface if they were present. More precisely, for $n \leq n_{\text{out}}$, from the construction of $\gamma_{[0,n]}(P)$ and the parity constraint in the definition of $\mathcal{P}_\Omega^\xi$, it follows that $C_{\gamma,n} \cup \mathcal{P}_\Omega^\xi$ consists exactly of those configurations satisfying both of the following properties:

1. $\gamma_t \in P$ for all $1 \leq t \leq n \vee (n_{\text{out}} - 1)$;
2. $e \notin P$ for all other edges $e$ which share a vertex with two edges in the path $\gamma_{[0,n]}$ that do not form a NE/SW pair (such as the edges drawn in red in Figure 2.6) — indeed, such edges would have taken priority according to the NE/SW rule.

In particular, if these two requirements are contradictory or the second is impossible, $\gamma_{[0,n]}$ does not respect the NE/SW rule and $C_{\gamma,n}$ is empty. Aside from this degenerate case, the set of edges which are not constrained coincides with $\mathcal{E}_{\Omega \setminus \tilde{\gamma}_{[0,n]}}$, where $\tilde{\gamma}_{[0,n]}$ is the set of vertices $v \in \mathcal{V}^*$ adjacent to at least one of the fixed edges — see Figure 2.6.

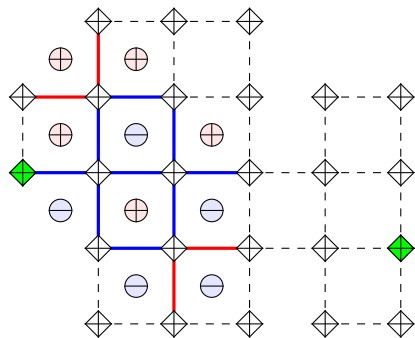

Figure 2.6: An example of $\tilde{\gamma}_{[0,n]}$ with the same $\gamma$ as in the previous figures and $n = 9$. The edges which are always included (resp. excluded) for all $P \in C_{\gamma,n}$ are drawn in blue (resp. red). The set $\tilde{\gamma}_{[0,n]}$ can also be understood as the set of vertices whose relative spin is fixed by the presence of the interface up to time $n$, and the vertices in $\tilde{\gamma}_{[0,n]}$ (not necessarily in $\Omega$) are shown as "$\oplus$" or "$\ominus$" according to how their relative spin is fixed.

As a result, we make the following observation, which will be useful in Section 3.3.

**Lemma 2.3.** *For each Peierls interface $\gamma$ as in Definition 2.2, integer $n \in \mathbb{Z}_{\geq 0}$, and boundary condition $\xi = (h_{\mathrm{in}}, h)$, there exists a set $\tilde{\gamma}_{[0,n]} \subset \mathcal{V}^*$ depending only on $\gamma_{[0,n]}$ such that the assignment*

$$P = \hat{P} \cup \{\gamma_1, \ldots, \gamma_n\} \tag{2.9}$$

*is a one-to-one correspondence between $P \in \mathcal{P}_\Omega^\xi$ and $\hat{P} \in \mathcal{P}_{\Omega \setminus \tilde{\gamma}_{[0,n]}}^{\vec{\gamma}_n, h}$.*

Now, assume that for each $h_{\mathrm{out}}$ such that $\xi = (h_{\mathrm{in}}, h_{\mathrm{out}})$ is admissible, the argument of $W$ is constant on $\mathcal{P}_\Omega^\xi$. Then, we may define a family of probability measures

$$\mathbb{P}_W^{\vec{\gamma}_0, h_{\mathrm{out}}}[\gamma_{[0,n]}] := \frac{F_W(\gamma_{[0,n]}; h_{\mathrm{out}})}{F_W(\gamma_{[0,0]}; h_{\mathrm{out}})}, \qquad n \in \mathbb{Z}_{\geq 0}. \tag{2.10}$$

For example, when $|W(P)| = \prod_{e \in P} x_e$, this is the distribution of the interface associated with the Ising model with Dobrushin boundary conditions, as defined in Section 2.1.

First step analysis of the interface shows that any function of the form (2.8) gives rise to a martingale (up to a suitable stopping time).

**Proposition 2.4.** *Fix $h = (f, e) \in \mathcal{V}_\Omega^c$ and a boundary condition[5] $\xi = (h_{\mathrm{in}}, h_{\mathrm{out}})$. The process*

$$M_W(h)[\gamma_{[0,n]}] := \frac{F_W(\gamma_{[0,n]}; h)}{F_W(\gamma_{[0,n]}; h_{\mathrm{out}})}, \qquad n \leq n_{\mathrm{out}}, \tag{2.11}$$

*is a local martingale with respect to the natural filtration generated by $\gamma$.*

---

[5]Here, we do not require $\xi$ to be admissible — while for relating this to the spin-Ising model we should, for models on contour configurations this makes sense for any boundary condition.

*Proof.* Let $n < n_{\text{out}}$. Note that the $(n+1)$st step of $\gamma$ satisfies

$$\mathbb{P}_W^\xi[\gamma_{n+1}|\gamma_{[0,n]}] = \frac{F_W(\gamma_{[0,n+1]};h_{\text{out}})}{F_W(\gamma_{[0,n]};h_{\text{out}})}. \tag{2.12}$$

Given $\gamma_{[0,n]}$, consider the possible dual edges $e_+ \in \mathcal{E}$ for which $\gamma_+ := \gamma_{[0,n]} \cup \{e_+\}$ has positive probability (i.e., the possible continuations of the Peierls interface). As the different events $C_{\gamma+,n+1}$ are disjoint and their union is $C_{\gamma,n}$, the definition (2.8) of $F_W$ gives

$$\begin{aligned}
\sum_{e_+} F_W(\gamma_+;h) &= \sum_{P \in \mathcal{P}_\Omega^\varpi} \sum_{e_+} \mathbb{1}_{C_{\gamma+,n+1}}(P)\,W^\varpi(P) \\
&= \sum_{P \in \mathcal{P}_\Omega^\varpi} \mathbb{1}_{C_{\gamma,n}}(P)\,W^\varpi(P) = F_W(\gamma_{[0,n]};h),
\end{aligned} \tag{2.13}$$

where $\varpi = (h_{\text{in}}, h)$. Combining Equation (2.13) with Equation (2.12) yields

$$\begin{aligned}
\mathbb{E}_W^\xi\Big[M_W(h)[\gamma_{[0,n+1]}] \,\Big|\, \gamma_{[0,n]}\Big] &= \sum_{e_+} \mathbb{P}_W^\xi[\gamma_{n+1} = e_+ \mid \gamma_{[0,n]}] \, \frac{F_W(\gamma_+;h)}{F_W(\gamma_+;h_{\text{out}})} \\
&\stackrel{(2.12)}{=} \frac{1}{F_W(\gamma_{[0,n]};h_{\text{out}})} \sum_{e_+} F_W(\gamma_+;h) \\
&\stackrel{(2.13)}{=} M_W(h)[\gamma_{[0,n]}].
\end{aligned} \tag{2.14}$$

This shows the martingale property for one step, which is sufficient to conclude. $\qquad\square$

Note that even for a fixed distribution of the interface process there are many possible functions $W$ with the needed properties, defining different martingale observables. For example, the original one used by Smirnov [Smi06] and Chelkak & Smirnov [CS12], further elaborated especially by Izyurov [Izy17], is a linear combination with two different values of $h$ sharing the same edge, as well as involving a specific choice of the functions $W$ with a sign depending on the winding of the contours. Appropriate choices are important to actually get a meromorphic scaling limit via discrete complex analysis techniques.

# 3    Grassmann integral representation of the planar Ising model

In this section, we review the representation of the planar (integrable) Ising model in terms of Grassmann variables, in particular introducing a specific choice of the functions $W$ which produces a version of the martingale observable used to study the interface process. In particular, we obtain an expression which is already known to have a suitable scaling limit.

## 3.1    Grassmann field, aka lattice fermion

Fix finite $\Omega \subset \mathcal{V}^\bullet$ and consider the cluster graph $\mathscr{G}_\Omega^c = (\mathscr{V}_\Omega^c, \mathscr{E}_\Omega^c)$ associated to $\Omega$. Let $\Phi_\Omega$ be the complex *Grassmann algebra* with a basis $\varphi_h$ indexed by $h \in \mathscr{V}_\Omega^c$, that is, $\Phi_\Omega$ is isomorphic to the exterior algebra of the vector space $V_\Omega := \mathrm{Span}_\mathbb{C}\{\varphi_h \mid h \in \mathscr{V}_\Omega^c\}$ obtained as the quotient of the tensor algebra of $V_\Omega$ by the relation $\varphi_h\varphi_{h'} = -\varphi_{h'}\varphi_h$ for

all $h, h' \in \mathscr{V}_\Omega^c$. For each face $f \in \mathcal{F}_\Omega$ we label the vertices of the associated complete four-graph $\bigoplus_f$ as $h_f^N, h_f^E, h_f^S, h_f^W \in \mathscr{V}_\Omega^c$, according to the direction of the associated edge (North, East, South, West) as seen from $f$. For brevity, we also denote the four associated Grassmann basis variables by

$$\varphi_f^N := \varphi_{h_f^N}, \qquad \varphi_f^E := \varphi_{h_f^E}, \qquad \varphi_f^S := \varphi_{h_f^S}, \qquad \varphi_f^W := \varphi_{h_f^W}.$$

Denote the subalgebra they generate by $\Phi_f$, and the associated Grassmann (Berezin) integral (see, e.g., [DFMS97, Appendix 2.B]) as

$$\int \mathcal{D}[\Phi_f] \, (\cdot) := \int \mathrm{d}\varphi_f^W \, \mathrm{d}\varphi_f^S \, \mathrm{d}\varphi_f^E \, \mathrm{d}\varphi_f^N \, (\cdot) \tag{3.1}$$

(note the possibly counterintuitive ordering which is chosen to simplify a sign later). The Grassmann action functional is given by the interaction of Grassmann basis variables across all short edges. The action functional around one face $f \in \mathcal{F}_\Omega$ is

$$\mathcal{S}_0(\Phi_f) := \varphi_f^E \varphi_f^W + \varphi_f^N \varphi_f^S + \varphi_f^N \varphi_f^E + \varphi_f^S \varphi_f^W + \varphi_f^S \varphi_f^E + \varphi_f^W \varphi_f^N. \tag{3.2}$$

Recalling that squares of Grassmann variables equal zero, and so $\exp(\mathcal{S}_0(\Phi_f)) = 1 + \mathcal{S}_0(\Phi_f)$, and noting that any Grassmann integral not involving all of the Grassmann variables exactly once vanishes, we obtain

$$\int \mathcal{D}[\Phi_f] \, e^{\mathcal{S}_0(\Phi_f)} = 1 = \int \mathcal{D}[\Phi_f] \, \varphi_f^N \, \varphi_f^E \, \varphi_f^S \, \varphi_f^W, \tag{3.3}$$

$$\int \mathcal{D}[\Phi_f] \, \varphi_f^N \varphi_f^E \varphi_f^S \varphi_f^W e^{\mathcal{S}_0(\Phi_f)} = 1 = \int \mathcal{D}[\Phi_f] \, \varphi_f^N \varphi_f^E e^{\mathcal{S}_0(\Phi_f)} \int \mathcal{D}[\Phi_f] \varphi_f^S \varphi_f^W e^{\mathcal{S}_0(\Phi_f)} \tag{3.4}$$

(cf. [DFMS97, Equations (2.219) & (2.225)]). The interaction across long edges is given by

$$E_e := \begin{cases} \varphi_f^E \, \varphi_{f+1}^W, & e = \{f, f+1\} \subset \mathcal{E}_\Omega, \\ \varphi_f^N \, \varphi_{f+\mathrm{i}}^S, & e = \{f, f+\mathrm{i}\} \subset \mathcal{E}_\Omega, \end{cases} \tag{3.5}$$

where $f$ denotes the midpoint of the face $f$, i.e., a point in dual lattice $(\mathbb{Z} + \frac{1}{2}) + \mathrm{i}(\mathbb{Z} + \frac{1}{2})$. We use this to define a Grassmann polynomial on $\Omega$, sometimes called an action functional,

$$\mathcal{S}_{\underline{x}}(\Phi_\Omega) := \sum_{e \in \mathcal{E}_\Omega} E_e \, x_e + \sum_{f \in \mathcal{F}_\Omega} \mathcal{S}_0(\Phi_f), \tag{3.6}$$

where $\underline{x} = (x_e)_{e \in \mathcal{E}_\Omega} \in [0, \infty)^{\mathcal{E}_\Omega}$ are arbitrary edge weights. Defining a Grassmann integral over the entire algebra via (3.1) by

$$\int \mathcal{D}[\Phi_\Omega] \, (\cdot) := \int \prod_{f \in \mathcal{F}_\Omega} \mathcal{D}[\Phi_f] \, (\cdot) = \int \prod_{f \in \mathcal{F}_\Omega} \mathrm{d}\varphi_f^W \, \mathrm{d}\varphi_f^S \, \mathrm{d}\varphi_f^E \, \mathrm{d}\varphi_f^N \, (\cdot)$$

(note that the order of the faces is not important), the key feature of this object is that it reproduces the partition function with fixed (plus) boundary conditions as

$$Z_{\Omega, \underline{x}}^+ = \int \mathcal{D}[\Phi_\Omega] \, e^{\mathcal{S}_{\underline{x}}(\Phi_\Omega)}, \tag{3.7}$$

since the integral is given by the same Pfaffian as in Kasteleyn's solution of the Ising model [Kas63] (see also [Sam80], and [CCK17] for the case with generic edge weights)[6]. This is closely related to the following identity, which will be the key to relating these integrals to other quantities of interest:

**Lemma 3.1.** *For any $X \subset \mathcal{E}_\Omega$, we have*

$$\int \mathcal{D}[\Phi_\Omega]\Big( \prod_{e \in X} E_e \Big) e^{\mathcal{S}_0(\Phi_\Omega)} = \mathbb{1}_{\mathcal{P}_\Omega}(X). \tag{3.8}$$

Note that this is in fact equivalent to one of the intermediate steps in Kasteleyn's solution, but rather than entering into the technical details needed to verify this, we instead derive it from the fact that Equation (3.7) holds for all values of the edge weights.

*Proof.* From Equation (2.4), we obtain

$$Z^+_{\Omega,\underline{x}} = \sum_{P \in \mathcal{P}_\Omega} \prod_{e \in P} x_e = \sum_{X \subset \mathcal{E}_\Omega} \mathbb{1}_{\mathcal{P}_\Omega}(X) \prod_{e \in X} x_e.$$

On the other hand, using Equation (3.7) and expanding the first term in the exponential in Equation (3.6) as a product of terms of the form $\exp(x_e E_e) = 1 + x_e E_e$, we obtain

$$Z^+_{\Omega,\underline{x}} = \int \mathcal{D}[\Phi_\Omega]\, e^{\mathcal{S}_{\underline{x}}(\Phi_\Omega)} = \sum_{X \subset \mathcal{E}_\Omega} \Big( \prod_{e \in X} x_e \Big) \int \mathcal{D}[\Phi_\Omega]\Big( \prod_{e \in X} E_e \Big) e^{\mathcal{S}_0(\Phi_\Omega)}.$$

Since the edge weights may be arbitrary, these two expressions are equal as polynomials, and equating the coefficients gives Equation (3.8). $\qquad\square$

We will find it convenient to define a Berezin measure related to the expression for the partition function in Equation (3.7) by

$$\langle \mathcal{O} \rangle^+_{\Omega,\underline{x}} := \frac{\int \mathcal{D}[\Phi_\Omega]\, \mathcal{O}(\Phi_\Omega)\, e^{\mathcal{S}_{\underline{x}}(\Phi_\Omega)}}{\int \mathcal{D}[\Phi_\Omega]\, e^{\mathcal{S}_{\underline{x}}(\Phi_\Omega)}} = \frac{1}{Z^+_{\Omega,\underline{x}}} \int \mathcal{D}[\Phi_\Omega]\, \mathcal{O}(\Phi_\Omega)\, e^{\mathcal{S}_{\underline{x}}(\Phi_\Omega)}, \tag{3.9}$$

where $\mathcal{O} = \mathcal{O}(\Phi_\Omega)$ is a function of the Grassmann variables (i.e., a generic element of the Grassmann algebra).

## 3.2 Grassmann integrals and boundary conditions

As we will now show, similar expressions are available for the sets of non-even contours involved in Dobrushin and alternating boundary conditions. (This could also be done in terms of the relationship to the dimer model on the cluster graph, but we will instead give a derivation in terms of manipulations of Grassmann integrals, which also produces some formulas which will be helpful later.)

Given a finite set $\Omega \subset \mathcal{V}^\bullet$, an integer $N \in \mathbb{Z}_{\geq 0}$, and distinct vertices (half-edges) $\xi = (h_1, h_2, \ldots, h_{2N})$ in $\mathscr{V}^c_\Omega$, let

$$W^\xi_{\Omega,\underline{x}}(X) := \int \mathcal{D}[\Phi_\Omega]\, \varphi_{h_1} \varphi_{h_2} \cdots \varphi_{h_{2N}} \Big( \prod_{e \in X} x_e E_e \Big) e^{\mathcal{S}_0(\Phi_\Omega)}, \qquad X \subset \mathcal{E}_\Omega. \tag{3.10}$$

---

[6]Equation (3.7) and similar formulae often appear with an ambiguous sign, but the fact that the signs of the two expressions match can be checked by setting $\underline{x} = 0$ and using Equation (3.3).

In Corollary 3.5, we relate similar Grassmann integrals to the partition function $Z_{\Omega,\underline{x}}^{\xi}$.

**Lemma 3.2.** *For any $X \subset \mathcal{E}_\Omega$, we have*

$$\sum_{\substack{P \in \mathcal{P}_\Omega^\xi \\ P \supset X}} W_{\Omega,\underline{x}}^\xi(P) = \Big( \prod_{e \in X} x_e \frac{\partial}{\partial x_e} \Big) \sum_{P \in \mathcal{P}_\Omega^\xi} W_{\Omega,\underline{x}}^\xi(P)$$

$$= \int \mathcal{D}[\Phi_\Omega]\, \varphi_{h_1}\, \varphi_{h_2} \cdots \varphi_{h_{2N}} \Big( \prod_{e \in X} x_e E_e \Big) e^{\mathcal{S}_{\underline{x}}(\Phi_\Omega)}. \tag{3.11}$$

*Proof.* Fix $X \subset \mathcal{E}_\Omega$. The first line of (3.11) is evident from (3.10). It remains to prove the second line of (3.11). By expanding the first term in the exponential in Equation (3.6) as a product of terms of the form $\exp(x_e E_e) = 1 + x_e E_e$, and noting that inside the Grassmann integral all terms which do not correspond to contour configurations $P \in \mathcal{P}_\Omega^\xi$ vanish since they cannot be written without repetitions of Grassmann variables, we obtain

$$\int \mathcal{D}[\Phi_\Omega]\, \varphi_{h_1}\, \varphi_{h_2} \cdots \varphi_{h_{2N}} \Big( \prod_{e \in X} x_e E_e \Big) e^{\mathcal{S}_{\underline{x}}(\Phi_\Omega)}$$

$$= \sum_{\substack{P \in \mathcal{P}_\Omega^\xi \\ P \supset X}} \int \mathcal{D}[\Phi_\Omega]\, \varphi_{h_1}\, \varphi_{h_2} \cdots \varphi_{h_{2N}} \Big( \prod_{e \in P} x_e E_e \Big) e^{\mathcal{S}_0(\Phi_\Omega)}.$$

The claim now follows by recognizing the definition (3.10) on the second line. $\qquad \square$

Among other things, this gives an almost explicit formula for $W_{\Omega,\underline{x}}$ which connects it to the distribution of contour configurations in the Ising model and to Section 2.3.

**Proposition 3.3.** *For any admissible boundary condition $\xi = (h_1, h_2, \ldots, h_{2N})$, we have*

$$W_{\Omega,\underline{x}}^\xi(P) = \pm \prod_{e \in P} x_e, \qquad \text{for all } P \in \mathcal{P}_\Omega^\xi, \tag{3.12}$$

*with the same sign for all $P \in \mathcal{P}_\Omega^\xi$.*

*Proof.* Since $\xi$ is admissible, there exist $\Omega_o \supset \Omega$ and $P_o \subset \mathcal{E}_{\Omega_o \setminus \Omega}$ with $|\Omega_o| < \infty$ such that $P \cup P_o \in \mathcal{P}_{\Omega_o}^\emptyset$ for all $P \in \mathcal{P}_\Omega^\xi$. Then evidently,

$$\varphi_{h_1}\, \varphi_{h_2} \cdots \varphi_{h_{2N}} = \pm \int \Big( \prod_{f \in \mathcal{F}_{\Omega_o} \setminus \mathcal{F}_\Omega} \mathcal{D}[\Phi_f]\, e^{\mathcal{S}_0(\Phi_f)} \Big) \Big( \prod_{e \in P_o} E_e \Big). \tag{3.13}$$

Set $x_e = 1$ for dual edges $e \in \mathcal{E}_{\Omega_o} \setminus \mathcal{E}_\Omega$. Then, we obtain

$$W_{\Omega,\underline{x}}^\xi(P) = \int \mathcal{D}[\Phi_\Omega]\, \varphi_{h_1}\, \varphi_{h_2} \cdots \varphi_{h_{2N}} \Big( \prod_{e \in P} x_e E_e \Big) e^{\mathcal{S}_0(\Phi_\Omega)} \qquad \text{[by (3.10)]}$$

$$= \pm \int \mathcal{D}[\Phi_{\Omega_o}] \Big( \prod_{e \in P_o \cup P} x_e E_e \Big) e^{\mathcal{S}_0(\Phi_\Omega)}$$

$$= \pm \prod_{e \in P} x_e, \qquad P \in \mathcal{P}_\Omega^\xi, \qquad \text{[by Lemma 3.1]}$$

where the sign on both lines equals that in (3.13), so it is indeed independent of $P \in \mathcal{P}_\Omega^\xi$. $\quad \square$

**Remark 3.4.** Without assuming that $\xi$ is admissible, Equation (3.12) still holds, but we lose control over the dependence of the sign on $P$.

Combining this with the $X = \emptyset$ case of Lemma 3.2 gives an expression for the partition function, up to a sign:

**Corollary 3.5.** *For any admissible boundary condition* $\xi = (h_1, h_2, \ldots, h_{2N})$,

$$Z^\xi_{\Omega,\underline{x}} = \left| \int \mathcal{D}[\Phi_\Omega] \, \varphi_{h_1} \, \varphi_{h_2} \, \cdots \, \varphi_{h_{2N}} \, e^{\mathcal{S}_{\underline{x}}(\Phi_\Omega)} \right|. \tag{3.14}$$

Note that in this case, unlike in the case of uniform boundary conditions, the Grassmann integral does not have a fixed sign; in particular it is odd under permutations of the order of the Grassmann variables inserted in the integral, and thus also depends non-trivially on the order of the elements of $\xi$, which is why we have defined it as an ordered tuple.

## 3.3 Grassmann integral representation of the martingale observable

From Proposition 3.3, we see that $W$ defined in Equation (3.10) in Section 3.2 has the properties used in Section 2.3 to construct a martingale observable with respect to the exploration process. To be precise, for admissible boundary condition $\xi = (h_{\mathrm{in}}, h_{\mathrm{out}})$, defining $F_W$ as in Equation (2.8), that is,

$$F_W(\gamma_{[0,n]}; h) := \sum_{P \in \mathcal{P}^{h_{\mathrm{in}}, h}_\Omega} \mathbb{1}_{C_{\gamma,n}}(P) \, W^{h_{\mathrm{in}}, h}_{\Omega,\underline{x}}(P), \tag{3.15}$$

where $C_{\gamma,n} := \{P \subset \mathcal{E}_\Omega \mid \gamma_{[0,n]} = \gamma_{[0,n]}(P)\}$, for each $h \in \mathscr{V}^c_\Omega \setminus \{h_{\mathrm{in}}\}$ the ratio in Equation (2.11), that is,

$$M_W(h)[\gamma_{[0,n]}] := \frac{F_W(\gamma_{[0,n]}; h)}{F_W(\gamma_{[0,n]}; h_{\mathrm{out}})},$$

is a (local) martingale (up to time $n = n_{\mathrm{out}}$) with respect to the exploration process associated with the Gibbs measure for the Ising model with Dobrushin boundary conditions specified by $\xi$.

Using Lemma 2.3, we obtain the following identity, which makes it possible to write $F$ more compactly as a Grassmann (Berezin) integral.

**Lemma 3.6.** *For each half-edge* $h \in \mathscr{V}^c_\Omega \setminus \{h_{\mathrm{in}}\}$, *each Peierls interface* $\gamma$ *as in Definition 2.2 with* $\vec{\gamma}_0 := h_{\mathrm{in}}$, *and each integer* $n \in \mathbb{Z}_{\geq 0}$, *there is a one-to-one correspondence between contours* $P \in \mathcal{P}^{\vec{\gamma}_0, h}_\Omega \cap C_{\gamma,n}$ *and* $\hat{P}(n) \in \mathcal{P}^{\vec{\gamma}_n, h}$ *such that*

$$W^{\vec{\gamma}_0, h}_{\Omega,\underline{x}}(P) = \pm \Big( \prod_{k=1}^n x_{\gamma_k} \Big) W^{\vec{\gamma}_n, h}_{\Omega \setminus \tilde{\gamma}_{[0,n]}, \underline{x}}(\hat{P}(n)), \tag{3.16}$$

*where* $\gamma$ *is the Peierls interface,* $\tilde{\gamma}_{[0,n]}$ *is the set in Lemma 2.3, and the sign is determined by the interface up to time* $n$; *in particular, it is independent of* $h$ *and* $P$.

*Proof.* From Lemma 2.3, we see that there is a unique contour configuration $\hat{P}(n)$ such that $P = \hat{P}(n) \cup \{\gamma_1, \ldots, \gamma_n\}$. Splitting $E_{\gamma_n}$ from Equation (3.5) as a product of $\varphi_{(f(\vec{\gamma}_{n-1}), \gamma_n)}$ and $\varphi_{\vec{\gamma}_n}$, we obtain

$$W_{\Omega, \underline{x}}^{\vec{\gamma}_0, h}(P) \tag{3.17}$$

$$= \int \mathcal{D}[\Phi_\Omega] \, \varphi_{\vec{\gamma}_0} \, \varphi_h \Big( \prod_{e \in \hat{P}(n) \cup \{\gamma_1, \ldots, \gamma_n\}} x_e E_e \Big) e^{\mathcal{S}_0(\Phi_\Omega)}$$

$$= \pm \int \mathcal{D}[\Phi_\Omega] \underbrace{\left( \varphi_{\vec{\gamma}_0} \Big( \prod_{k=1}^{n-1} x_{\gamma_k} E_{\gamma_k} \Big) x_{\gamma_n} \varphi_{(f(\vec{\gamma}_{n-1}), \gamma_n)} \right)}_{(A)} \underbrace{\left( \varphi_{\vec{\gamma}_n} \varphi_h \Big( \prod_{e \in \hat{P}(n)} x_e E_e \Big) \right)}_{(B)} e^{\mathcal{S}_0(\Phi_\Omega)},$$

with the same sign $\pm$ as in $E_{\gamma_n} = \pm \varphi_{v(\vec{\gamma}_{n-1}, \gamma_n)} \varphi_{v(\vec{\gamma}_n, \gamma_n)}$, which depends only on $\gamma_{[0,n]}$.

We will separate the two parts (A) & (B) in the integral into two disjoint Grassmann integrals. Such a separation is not completely trivial, since the sets $\mathcal{F}_{\Omega \setminus \tilde{\gamma}_{[0,n]}}$ and $\mathcal{F}_{\tilde{\gamma}_{[0,n]}}$ of faces respectively incident to the dual edges in $\mathcal{E}_{\Omega \setminus \tilde{\gamma}_{[0,n]}}$ and $\mathcal{E}_{\tilde{\gamma}_{[0,n]}}$ have a non-trivial intersection, consisting of faces $f$ which we divide into three types (see Figure 2.6):

1. $f$ that are not incident to dual edges belonging to the Peierls interface $\gamma_{[0,n]}$;
2. $f$ that are incident to $\gamma_{[0,n]}$ but not incident to $\hat{P}(n)$;
3. $f$ incident both to dual edges belonging to the Peierls interface $\gamma_{[0,n]}$ and to $\hat{P}(n)$.

We can duplicate the integrals over these faces as follows.

In Cases 1 and 2, we may insert a factor (3.2) equaling one to the Grassmann integral associated to the corresponding face:

$$1 = \int \mathcal{D}[\Phi_f] \, e^{\mathcal{S}_0(\Phi_f)}$$

which is grouped with (A) in Case 1 and with (B) in Case 2.

In Case 3, the parity constraints and the disambiguation rule for the Peierls interface (resolving the self-touchings of the interface in the North-East and South-West directions) guarantee that the relevant part of the integral contains all four associated Grassmann variables, and so has the form

$$\int \mathcal{D}[\Phi_f] \varphi_f^N \varphi_f^E \varphi_f^S \varphi_f^W e^{\mathcal{S}_0(\Phi_f)} = 1 = \int \mathcal{D}[\Phi_f] \varphi_f^N \varphi_f^E e^{\mathcal{S}_0(\Phi_f)} \int \mathcal{D}[\Phi_f] \varphi_f^S \varphi_f^W e^{\mathcal{S}_0(\Phi_f)}$$

using Equation (3.4), up to a reordering which does not change the sign of the final result. It now follows that

$$(3.17) = \pm \Big( \prod_{k=1}^{n} x_{\gamma_k} \Big) \underbrace{\int \mathcal{D}[\Phi_{\tilde{\gamma}_{[0,n]}}] \varphi_{v_{h_{\mathrm{in}}}} \varphi_{v(\vec{\gamma}_{n-1}, \gamma_n)} \Big( \prod_{k=1}^{n-1} E_{\gamma_k} \Big) e^{\mathcal{S}_0(\Phi_{\tilde{\gamma}_{[0,n]}})}}_{= \pm 1}$$

$$\times \underbrace{\int \mathcal{D}[\Phi_{\Omega \setminus \tilde{\gamma}_{[0,n]}}] \varphi_{v(\vec{\gamma}_n, \gamma_n)} \varphi_{v_h} \Big( \prod_{e \in \hat{P}(n)} x_e E_e \Big) e^{\mathcal{S}_0(\Phi_{\Omega \setminus \tilde{\gamma}_{[0,n]}})}}_{= W_{\Omega \setminus \tilde{\gamma}_{[0,n]}, \underline{x}}^{\vec{\gamma}_n, f}(\hat{P}(n))} \tag{3.18}$$

where the sign "±" is the same as the one in Equation (3.17). Note that the first integral in (3.18) only depends on $\gamma_{[0,n]}$. Hence, we obtain the asserted identity (3.16), with a different sign compared to Equation (3.17) which, however, still depends only on $\gamma_{[0,n]}$. $\square$

This makes it possible to replace the sum in Equation (3.15) with a sum over $\mathcal{P}^{\vec{\gamma}_n,h}_{\Omega\setminus\tilde{\gamma}_{[0,n]}}$, and Lemma 3.2 lets us rewrite this sum in terms of a Grassmann integral as

$$F_W(\gamma_{[0,n]};h) = \pm\Big(\prod_{k=1}^{n} x_{\gamma_k}\Big) \int \mathcal{D}[\Phi_{\Omega\setminus\tilde{\gamma}_{[0,n]}}]\, \varphi_{v(\vec{\gamma}_n,\gamma_n)}\, \varphi_{v_h}\Big(\prod_{e\in\hat{P}(n)} x_e E_e\Big) e^{\mathcal{S}_{\underline{x}}^{\pm}(\Phi_{\Omega\setminus\tilde{\gamma}_{[0,n]}})}. \quad (3.19)$$

As a result, the martingale observable can be expressed in terms of the Berezin measure introduced in Equation (3.9):

**Proposition 3.7.** *For each admissible boundary condition $\xi = (h_{\mathrm{in}}, h_{\mathrm{out}})$ and half-edge $h \in \mathscr{V}_\Omega^c \setminus \{h_{\mathrm{in}}\}$, we have*

$$M_W(h)[\gamma_{[0,n]}] = \frac{\big\langle \varphi_{\vec{\gamma}_n}\, \varphi_h \big\rangle^+_{\Omega\setminus\tilde{\gamma}_{[0,n]},\underline{x}}}{\big\langle \varphi_{\vec{\gamma}_n}\, \varphi_{h_{\mathrm{out}}} \big\rangle^+_{\Omega\setminus\tilde{\gamma}_{[0,n]},\underline{x}}} \quad (3.20)$$

*where $\gamma$ is the Peierls interface as in Definition 2.2.*

To proceed with the proof of convergence of the interface, at this point one would need to discuss the convergence of this ratio in the scaling limit, which follows by relating it to an object with suitable discrete holomorphicity properties. It is presumably the case that $M_W$ is sufficiently similar to that of [Smi06, CI13, Izy15] to repeat the argument used there, but this involves carefully studying the sign of $W$ (cf. Remark 3.4). As a simpler alternative, let us note that these are the same Berezin expectation values studied in [CCK17], where they are also expressed in terms of s-holomorphic objects with appropriate boundary conditions.

# 4 Martingale observable for non-integrable Ising models

We now turn to the version of the Ising model with formal Hamiltonian (1.1), which we define more precisely in Equation (4.1) and Lemma 4.1. We use the following quantities.

- Let $U\colon 2^{\mathcal{E}} \to \mathbb{R}$ be a finite-range, translation-invariant interaction, i.e., such that there exists a radius $R \in (0,\infty)$ for which $\max\limits_{X\in\mathrm{supp}\,U} \mathrm{diam}\,X = R$, and $U(X + a) = U(X)$ for all $a \in \mathbb{Z} + \mathrm{i}\mathbb{Z}$ and $X \subset \mathcal{E}$.

- Fix a finite vertex set $\Omega \subset \mathcal{V}^\bullet$ and an admissible $\xi = (h_1, h_2, \ldots, h_{2N})$.

- Let $\Omega_o \supset \Omega$ and $P_o \subset \mathcal{E}_{\Omega_o\setminus\Omega}$ with $|\Omega_o| < \infty$ be such that $P \cup P_o \in \mathcal{P}_{\Omega_o}$ for all $P \in \mathcal{P}_\Omega^\xi$.

Given a spin configuration $\sigma \in \{\pm1\}^{\mathcal{V}^\bullet_\Omega}$, define $\epsilon_{e^\bullet}(\sigma) := \frac{1}{2}(1 - \sigma_{v^\bullet}\sigma_{w^\bullet})$ for edges $e^\bullet = \{v^\bullet, w^\bullet\}$ as before, with spins for vertices outside of $\Omega$ fixed according to $P_o$ by,

$$\sigma_{v^\bullet} = (-1)^{|K_{v^\bullet}\cap P|}, \qquad v^\bullet \in \Omega_o \setminus \Omega,$$

where $K_{v^\bullet}$ is the set of dual edges crossed by a simple path from $v^\bullet$ to $\mathbb{Z}^2 \setminus \Omega_o$ without using any edges in $\mathcal{E}_\Omega^\bullet$ (as usual, parity constraints imply that it is independent of the choice of path).

To define the general (non-integrable) model, we consider the Hamiltonian with boundary condition $P_o \in \mathcal{P}_{\Omega_o}^\xi$ defined by

$$H_{\Omega,J,\lambda}^{P_o}(\sigma) := 2J \sum_{e^\bullet \in \mathcal{E}_\Omega^\bullet} \epsilon_{e^\bullet}(\sigma) + \lambda \sum_{X \subset \mathcal{E}_\Omega} U(X) \prod_{e \in X} \epsilon_{e^\bullet}(\sigma), \tag{4.1}$$

where $\lambda \in \mathbb{R}$ is a parameter controlling the strength of the added non-planar or multi-spin interaction (which should be thought of as fixed but small compared to $J$).

The Hamiltonian (4.1) is equivalent to that of Equation (1.1), which is the form used in [GGM12, AGG21, AGG23], since any even local function of the spins can be written in either form up to a constant term. The relationship can be made explicit as follows.

**Lemma 4.1.** *Suppose that*

$$U(X) = (-2)^{|X|} \sum_{\substack{Y^\bullet \subset \mathcal{E}_\Omega^\bullet \\ Y^\bullet \supset X^\bullet}} V(Y^\bullet), \qquad X \subset \mathcal{E}_\Omega,$$

*where $X \leftrightarrow X^\bullet$ is the bijection between dual edges in $X \subset \mathcal{E}_\Omega$ and edges in $X^\bullet \subset \mathcal{E}_\Omega^\bullet$. The Gibbs measures associated to the two Hamiltonians (1.1) and (4.1) agree in the sense that*

$$\frac{e^{-\beta H_{\Omega,J,\lambda}^\bullet(\sigma)}}{\sum_\sigma \exp(-\beta H_{\Omega,J,\lambda}^\bullet(\sigma))} = \frac{e^{-\beta H_{\Omega,J,\lambda}(\sigma)}}{\sum_\sigma \exp(-\beta H_{\Omega,J,\lambda}(\sigma))},$$

*where*

$$H_{\Omega,J,\lambda}^\bullet(\sigma) := -J \sum_{\{v^\bullet,w^\bullet\} \in \mathcal{E}^\bullet} \sigma_{v^\bullet} \sigma_{w^\bullet} + \lambda \sum_{Y^\bullet \subset \mathcal{E}^\bullet} V(Y^\bullet) \prod_{\{v^\bullet,w^\bullet\} \in Y^\bullet} \sigma_{v^\bullet} \sigma_{w^\bullet},$$

$$H_{\Omega,J,\lambda}(\sigma) := 2J \sum_{e^\bullet \in \mathcal{E}_\Omega^\bullet} \epsilon_{e^\bullet}(\sigma) + \lambda \sum_{X \subset \mathcal{E}_\Omega} U(X) \prod_{e \in X} \epsilon_{e^\bullet}(\sigma).$$

*Proof.* Note that the first terms in the two Hamiltonians can be related as

$$2J \sum_{e^\bullet \in \mathcal{E}_\Omega^\bullet} \epsilon_{e^\bullet}(\sigma) = J \sum_{\{v^\bullet,w^\bullet\} \in \mathcal{E}_\Omega^\bullet} (1 - \sigma_{v^\bullet} \sigma_{w^\bullet}) = J |\mathcal{E}_\Omega^\bullet| - J \sum_{\{v^\bullet,w^\bullet\} \in \mathcal{E}_\Omega^\bullet} \sigma_{v^\bullet} \sigma_{w^\bullet}.$$

The constant factor $J |\mathcal{E}_\Omega^\bullet|$ does not affect the Gibbs measure. The second (potentially integrability-breaking) term in the two Hamiltonians can be related as follows:

$$\sum_{X \subset \mathcal{E}_\Omega} U(X) \prod_{e \in X} \epsilon_{e^\bullet}(\sigma) = \sum_{X \subset \mathcal{E}_\Omega} (-2)^{|X|} \sum_{\substack{Y^\bullet \subset \mathcal{E}_\Omega^\bullet \\ Y^\bullet \supset X^\bullet}} V(Y^\bullet) \, 2^{-|X|} \sum_{A \subset X} (-1)^{|A|} \prod_{\{v^\bullet,w^\bullet\} \in A^\bullet} \sigma_{v^\bullet} \sigma_{w^\bullet}$$

$$= \sum_{\substack{A,X,Y \subset \mathcal{E}_\Omega \\ A \subset X \\ X^\bullet \subset Y^\bullet}} (-1)^{|A|} (-1)^{|X|} V(Y^\bullet) \prod_{\{v^\bullet,w^\bullet\} \in A^\bullet} \sigma_{v^\bullet} \sigma_{w^\bullet}.$$

For a fixed set $A \subset \mathcal{E}_\Omega$, the coefficient of $\prod_{\{v^\bullet, w^\bullet\} \in A^\bullet} \sigma_{v^\bullet} \sigma_{w^\bullet}$ is

$$\sum_{\substack{X, Y \subset \mathcal{E}_\Omega \\ A \subset X \\ X^\bullet \subset Y^\bullet}} (-1)^{|A|} (-1)^{|X|} V(Y^\bullet) = V(A^\bullet) + \sum_{\substack{Y \subset \mathcal{E}_\Omega \\ A \subsetneq Y}} (-1)^{|A|} \sum_{\substack{X \subset \mathcal{E}_\Omega \\ A \subset X \\ X^\bullet \subset Y^\bullet}} (-1)^{|X|} = V(A^\bullet),$$

because the binomial formula shows that

$$\sum_{\substack{X \subset \mathcal{E}_\Omega \\ A \subset X \\ X^\bullet \subset Y^\bullet}} (-1)^{|X|} = \sum_{k=0}^{|Y \setminus A|} (-1)^{|A|+k} \binom{|Y \setminus A|}{k} = 0.$$

Hence, we conclude that

$$\sum_{X \subset \mathcal{E}_\Omega} U(X) \prod_{e \in X} \epsilon_{e^\bullet}(\sigma) = \sum_{A \subset \mathcal{E}_\Omega} V(A^\bullet) \prod_{\{v^\bullet, w^\bullet\} \in A^\bullet} \sigma_{v^\bullet} \sigma_{w^\bullet},$$

which finishes the proof. $\qquad\square$

The Hamiltonian (4.1), like the planar one, can be written as a function of the low-temperature contour configuration $P(\sigma) := \{e \in \mathcal{E}_\Omega \mid \epsilon_{e^\bullet}(\sigma) = 1\}$, which as a result has the distribution

$$\mathbb{P}^{P_o}_{\Omega, x, \beta, \lambda}[P] := \frac{1}{Z^{P_o}_{\Omega, x, \beta, \lambda}} x^{|P|} \prod_{X \subset P} e^{-\beta\lambda U^{P_o}(X)}, \qquad P \in \mathcal{P}^\xi_\Omega, \tag{4.2}$$

corresponding to the Gibbs measure of Equation (4.1), with

$$U^{P_o}(X) := \sum_{Y \subset P_o} U(X \cup Y) \tag{4.3}$$

and partition function

$$Z^{P_o}_{\Omega, x, \beta, \lambda} = \sum_{P \in \mathcal{P}^\xi_\Omega} x^{|P|} \prod_{X \subset P} e^{-\beta\lambda U^{P_o}(X)},$$

where $x = e^{-2\beta J}$ is the (constant) edge weight and $\beta \in (0, \infty)$ is the inverse temperature. We will again focus on the case where $N = 1$ (Dobrushin boundary conditions, recall Figure 2.2), writing $\xi = (h_{\text{in}}, h_{\text{out}})$ although the generalization to $N \geq 2$ is straightforward.

The function $W^\xi_{\Omega, x}$ defined in (3.10) can be generalized[7] to the $\lambda \neq 0$ case as

$$\begin{aligned}
W^{\xi, P_o}_{\Omega, x, \beta, \lambda}(P) &:= W^\xi_{\Omega, x}(P) \prod_{X \subset P} e^{-\beta\lambda U^{P_o}(X)} \\
&= W^\xi_{\Omega, x}(P) \prod_{X \subset P} \Big( \prod_{Y \subset P_o} e^{-\beta\lambda U(X \cup Y)} \Big), \qquad P \subset \mathcal{E}_\Omega.
\end{aligned} \tag{4.4}$$

---

[7]From here on, we identify $x$ with the constant function in this and other quantities which depend on the edge weights.

This can then be used to define martingale observables as in Section 2.3, in particular since the second factor does not change the sign. More precisely, for the Peierls interface started at $\vec{\gamma}_0 = h_{\text{in}}$, the function defined in Equation (2.8),

$$F_W(\gamma_{[0,n]}; h) := \sum_{P \in \mathcal{P}_\Omega^{\vec{\gamma}_0, h}} \mathbb{1}_{C_{\gamma,n}}(P) \, W_{\Omega,x,\beta,\lambda}^{\vec{\gamma}_0, h, P_o}(P), \qquad h \in \mathscr{V}_\Omega^c, \tag{4.5}$$

where $C_{\gamma,n} := \{P \subset \mathcal{E}_\Omega \mid \gamma_{[0,n]} = \gamma_{[0,n]}(P)\}$, gives rise to the martingale observable

$$M_W(h)[\gamma_{[0,n]}] = \frac{F_W(\gamma_{[0,n]}; h)}{F_W(\gamma_{[0,n]}; h_{\text{out}})}, \qquad n \le n_{\text{out}}. \tag{4.6}$$

## 4.1 Grassmann integral representation

The observable (4.6) also admits a representation as a Grassmann integral, constructed as follows. For $P \subset \mathcal{E}_\Omega$, we denote the set of collections of non-overlapping subsets of $P$ by

$$\mathcal{H}(P) := \{\mathcal{Q} \subset 2^P \mid Q \cap Q' = \emptyset \text{ for all } Q, Q' \in \mathcal{Q}\}.$$

A collection $\mathcal{Y} \subset 2^P$ is said to be *overlap-connected* (o.c.) if the graph formed from $\mathcal{Y}$ by adding an edge for each pair $Y_1, Y_2 \in \mathcal{Y}$ with $Y_1 \cap Y_2 \neq \emptyset$ is connected.

**Proposition 4.2.** *For any $h_{\text{in}}, h \in \mathscr{V}_\Omega^c$ and any $P_o \subset \mathcal{E}_{\Omega_o \setminus \mathcal{E}_\Omega}$, we have*

$$\sum_{P \in \mathcal{P}_\Omega^{h_{\text{in}}, h}} W_{\Omega,x,\beta,\lambda}^{h_{\text{in}}, h, P_o}(P) = \int \mathcal{D}[\Phi_\Omega] \, \varphi_{h_{\text{in}}} \, \varphi_h \, e^{\mathcal{S}_x(\Phi_\Omega) + \mathcal{U}_{\beta,x,\lambda}^{P_o}(\Phi_\Omega)}. \tag{4.7}$$

*where $\mathcal{U}$ is a Grassmann polynomial of the form*

$$\mathcal{U}_{\beta,x,\lambda}^{P_o}(\Phi_\Omega) = \sum_{X \subset \mathcal{E}_\Omega} \overline{U}_{\beta,x,\lambda}^{P_o}(X) \Big( \prod_{e \in X} E_e \Big),$$

$$\overline{U}_{\beta,x,\lambda}^{P_o}(X) = x^{|X|} \sum_{\substack{\mathcal{Y} \subset 2^X \\ \mathcal{Y} \text{ o.c.} \\ \bigcup_{Y \in \mathcal{Y}} Y = X}} \prod_{Y \in \mathcal{Y}} \Big( \exp(-\beta \lambda U^{P_o}(Y)) - 1 \Big)$$

*Furthermore, whenever $X \subset \mathcal{E}_\Omega \cap \mathcal{E}_{\Omega'}$ and $P_o$ and $P_o'$ are valid boundary conditions for $\Omega$ and $\Omega'$ respectively, we have*

$$\overline{U}_{\beta,x,\lambda}^{P_o}(X) = \overline{U}_{\beta,x,\lambda}^{P_o'}(X) \quad \text{whenever} \quad \text{dist}(X, P_o \triangle P_o') > R, \tag{4.8}$$

*and there are $U$-dependent constants $C_U, c_U \in (0, \infty)$ such that for $|\beta\lambda| \le C_U$,*

$$|\overline{U}_{\beta,x,\lambda}^{P_o}(X)| \le C_U \, e^{-c_U \lambda T(X)}, \tag{4.9}$$

*where $T(X)$ denotes the minimum of $|Z|$ over connected sets $Z \subset \mathcal{E}$ such that $X \subset Z$.*

Note that we do not assume any particular relationship between $P_o$ and $h_{\text{in}}, h$.

*Proof.* Fix $P \subset \mathcal{E}_\Omega$. Expanding as a binomial, we can write

$$\prod_{X \subset P} \exp(-\beta\lambda U^{P_o}(X)) = \sum_{\mathcal{X} \subset 2^P} \prod_{X \in \mathcal{X}} \Big( \exp(-\beta\lambda U^{P_o}(X)) - 1 \Big)$$

$$= \sum_{\mathcal{Q} \in \mathcal{H}(P)} \prod_{Q \in \mathcal{Q}} \sum_{\substack{\mathcal{Y} \subset 2^Q \\ \mathcal{Y} \text{ o.c.} \\ \bigcup_{Y \in \mathcal{Y}} Y = Q}} \prod_{Y \in \mathcal{Y}} \Big( \exp(-\beta\lambda U^{P_o}(Y)) - 1 \Big)$$

$$=: \sum_{\mathcal{Q} \in \mathcal{H}(P)} \prod_{Q \in \mathcal{Q}} \tilde{U}^{P_o}_{\beta,\lambda}(Q). \tag{4.10}$$

Using the definition from Equation (4.4), we obtain

$$\sum_{P \in \mathcal{P}^{h_{\text{in}},h}_\Omega} W^{h_{\text{in}},h,P_o}_{\Omega,x,\beta,\lambda}(P) \tag{4.11}$$

$$= \sum_{P \in \mathcal{P}^{h_{\text{in}},h}_\Omega} W^{h_{\text{in}},h}_{\Omega,x}(P) \prod_{X \subset P} \exp(-\beta\lambda U^{P_o}(X)) \qquad \text{[by (4.4)]}$$

$$= \sum_{P \in \mathcal{P}^{h_{\text{in}},h}_\Omega} W^{h_{\text{in}},h}_{\Omega,x}(P) \sum_{\mathcal{Q} \in \mathcal{H}(P)} \prod_{Q \in \mathcal{Q}} \tilde{U}^{P_o}_{\beta,\lambda}(Q) \qquad \text{[by (4.10)]}$$

$$= \sum_{P \in \mathcal{P}^{h_{\text{in}},h}_\Omega} \Big( \prod_{Q \in \mathcal{Q}} \tilde{U}^{P_o}_{\beta,\lambda}(Q) \Big) \sum_{\substack{S \in \mathcal{P}^{f(h_{\text{in}}),h}_\Omega \\ S \supset \bigcup_{Q \in \mathcal{Q}} Q}} W^{f(h_{\text{in}}),h}_{\Omega,x}(S)$$

$$= \int \mathcal{D}[\Phi_\Omega] \, \varphi_{h_{\text{in}}} \, \varphi_h \sum_{\mathcal{Q} \in \mathcal{H}(\mathcal{E}_\Omega)} \Big( \prod_{Q \in \mathcal{Q}} \tilde{U}^{P_o}_{\beta,\lambda}(Q) \Big( \prod_{e \in Q} x E_e \Big) \Big) e^{\mathcal{S}_x(\Phi_\Omega)}. \qquad \text{[by Lemma 3.2]}$$

Now, since $\mathcal{H}(\mathcal{E}_\Omega)$ excludes only collections with overlapping sets, we may replace the sum over $\mathcal{H}(\mathcal{E}_\Omega)$ with the sum over all of $2^{\mathcal{E}_\Omega}$, since this yields only additional terms involving repeated Grassmann variables (which vanish). Doing so, we see that

$$\sum_{\mathcal{Q} \in \mathcal{H}(\mathcal{E}_\Omega)} \Big( \prod_{Q \in \mathcal{Q}} \tilde{U}^{P_o}_{\beta,\lambda}(Q) \Big( \prod_{e \in Q} x E_e \Big) \Big) = \sum_{\mathcal{Q} \in 2^{\mathcal{E}_\Omega}} \Big( \prod_{Q \in \mathcal{Q}} \tilde{U}^{P_o}_{\beta,\lambda}(Q) \Big( \prod_{e \in Q} x E_e \Big) \Big)$$

$$= \prod_{X \subset \mathcal{E}_\Omega} \Big( 1 + \tilde{U}^{P_o}_{\beta,\lambda}(X) \prod_{e \in X} x E_e \Big)$$

$$=: \prod_{X \subset \mathcal{E}_\Omega} \Big( 1 + \overline{U}^{P_o}_{\beta,x,\lambda}(X) \prod_{e \in X} E_e \Big) = \exp\big( \mathcal{U}^{P_o}_{\beta,x,\lambda}(\Phi_\Omega) \big),$$

and inserting this in Equation (4.11) gives the asserted identity (4.7).

To verify the partial independence property in Equation (4.8), first note from the definition in Equation (4.3) that $U^{P_o}(Y)$ has the same property for all $Y \subset X$, which together with Equation (4.10) implies the same for $\tilde{U}^{P_o}_{\beta,\lambda}(X)$ and so also for

$$\overline{U}^{P_o}_{\beta,x,\lambda}(X) = x^{|X|} \tilde{U}^{P_o}_{\beta,\lambda}(X) = \sum_{\substack{\mathcal{Y} \subset 2^X \\ \mathcal{Y} \text{ o.c.} \\ \bigcup_{Y \in \mathcal{Y}} Y = X}} \prod_{Y \in \mathcal{Y}} x^{|Y|} \Big( \exp(-\beta\lambda U^{P_o}(Y)) - 1 \Big). \tag{4.12}$$

The bound in Equation (4.8) follows by noting that the summand in (4.12) vanishes unless

$$\operatorname{diam} Y \le R, \text{ for all } Y \in \mathcal{Y}, \text{ and thus, also } R^2|\mathcal{Y}| \ge T(X).$$

It thus follows that the sum has no more than $R^{2|X|} \le \exp(2T(X)\log R)$ terms, each of which, noting that $x = e^{-2\beta J} \le 1$, is bounded by $((e-1)\beta\lambda)^{T(X)/R^2}$ as long as $|\beta\lambda| \le (\max_{Y \subset \mathcal{E}} |U(Y)|)^{-1}$. This gives the final bound (4.9). $\qquad \square$

We can use Proposition 4.2 to express the function $F_W$ defined in Equation (4.5) as a Grassmann integral as well. This leads to an expression for the martingale in Theorem 4.4.

**Lemma 4.3.** *Whenever $h \in \mathscr{V}_\Omega^c \notin \{\vec{\gamma}_0\}$, we have*

$$F_W(\gamma_{[0,n]}; h) = \pm x^n \int \mathcal{D}[\Phi_{\Omega \setminus \tilde{\gamma}_{[0,n]}}] \, \varphi_{\vec{\gamma}_0} \, \varphi_h \, e^{\mathcal{S}_x(\Phi_{\Omega \setminus \tilde{\gamma}_{[0,n]}}) + \mathcal{U}_{\beta,x,\lambda}^{\check{P}_o(n)}(\Phi_{\Omega \setminus \tilde{\gamma}_{[0,n]}})}, \qquad (4.13)$$

*where $\check{P}_o(n) := P_o \cup \{\gamma_1, \ldots, \gamma_n\}$, with a sign depending only on $\gamma_{[0,n]}$.*

*Proof.* Recalling the definition of $F_W$, with an appropriate sign, we obtain

$$
\begin{aligned}
& F_W(\gamma_{[0,n]}; h) \\
&= \sum_{P \in \mathcal{P}_\Omega^{\vec{\gamma}_0,h} \cap C_{\gamma,n}} W_{\Omega,x,\beta,\lambda}^{\vec{\gamma}_0,h,P_o}(P), && \text{[by Equation (4.5)]} \\
&= \sum_{P \in \mathcal{P}_\Omega^{\vec{\gamma}_0,h} \cap C_{\gamma,n}} W_{\Omega,x}^{h_{\text{in}},h}(P) \prod_{X \subset P} \prod_{Y \subset P_o} e^{-\beta\lambda U(X \cup Y)} && \text{[by Equation (4.4)]} \\
&= \pm x^n \sum_{\hat{P} \in \mathcal{P}_{\Omega \setminus \tilde{\gamma}_{[0,n]}}^{\vec{\gamma}_n,h}} W_{\Omega \setminus \tilde{\gamma}_{[0,n]},x}^{\vec{\gamma}_n,h}(\hat{P}) \prod_{X \subset \hat{P}} \prod_{Y \subset \check{P}_o(n)} e^{-\beta\lambda U(X \cup Y)} && \text{[by Lemma 3.6]} \\
&= \pm x^n \sum_{\hat{P} \in \mathcal{P}_{\Omega \setminus \tilde{\gamma}_{[0,n]}}^{\vec{\gamma}_n,h}} W_{\Omega \setminus \tilde{\gamma}_{[0,n]},x,\beta,\lambda}^{\vec{\gamma}_n,h,\check{P}_o(n)}(\hat{P}). && \text{[by Lemma 2.3]}
\end{aligned}
$$

The asserted identity (4.13) follows by applying Proposition 4.2 on $\Omega \setminus \tilde{\gamma}_{[0,n]}$. $\qquad \square$

Generalizing the Berezin measure introduced in Equation (3.9) to the general case,

$$\langle \mathcal{O} \rangle_{\Omega,x,\beta,\lambda}^{P_o} := \frac{\int \mathcal{D}[\Phi_\Omega] \, \mathcal{O} \, e^{\mathcal{S}_x(\Phi_\Omega) + \mathcal{U}_{\beta,x,\lambda}^{P_o}(\Phi_\Omega)}}{\int \mathcal{D}[\Phi_\Omega] \, e^{\mathcal{S}_x(\Phi_\Omega) + \mathcal{U}_{\beta,x,\lambda}^{P_o}(\Phi_\Omega)}}, \qquad (4.14)$$

for $\mathcal{O}$ in the Grassmann algebra $\Phi_\Omega$, we can rewrite $M_W$ as defined in Equation (4.6):

**Theorem 4.4.** *For any $h \in \mathscr{V}_\Omega^c \setminus \{\vec{\gamma}_0\}$, the process*

$$n \quad \longmapsto \quad \frac{\langle \varphi_{\vec{\gamma}_n} \varphi_h \rangle_{\Omega \setminus \tilde{\gamma}_{[0,n]},x,\beta,\lambda}^{\check{P}_o(n)}}{\langle \varphi_{\vec{\gamma}_n} \varphi_{h_{\text{out}}} \rangle_{\Omega \setminus \tilde{\gamma}_{[0,n]},x,\beta,\lambda}^{\check{P}_o(n)}}, \qquad n \le n_{\text{out}},$$

*is a local martingale with respect to the natural filtration generated by $\gamma$.*

*Proof.* Equation (4.6) defines a local martingale by Proposition 2.4. We have

$$M_W(h)[\gamma_{[0,n]}] = \frac{F_W(\gamma_{[0,n]};h)}{F_W(\gamma_{[0,n]};h_{\mathrm{out}})} = \frac{\langle \varphi_{\vec{\gamma}_n}\, \varphi_h \rangle^{\check{P}_o(n)}_{\Omega\setminus\tilde{\gamma}_{[0,n]},x,\beta,\lambda}}{\langle \varphi_{\vec{\gamma}_n}\, \varphi_{h_{\mathrm{out}}} \rangle^{\check{P}_o(n)}_{\Omega\setminus\tilde{\gamma}_{[0,n]},x,\beta,\lambda}}$$

by Lemma 4.3 and Equation (4.14). $\qquad\square$

## 4.2   Conjecture about the scaling limit

Let us now present a precise conjecture concerning the martingale observable, which would imply its convergence in a locally uniform fashion to the same scaling limit as in the well-studied planar case. We consider the non-integrable model with the Hamiltonian (4.1).

**Conjecture 4.5.** *There exists $\lambda_0 = \lambda_0(U) > 0$ such that for all $|\lambda| \le \lambda_0$, there exists $\beta^* = \beta^*(\lambda)$ such that for all finite $\Omega \subset \mathcal{V}^\bullet$ and for all $\xi = (h_1, h_2, \ldots, h_{2N})$ and associated external boundary conditions $P_o \subset \mathcal{E}_{\Omega_o\setminus\Omega}$, there exists a collection $\{\zeta^{P_o}_{\Omega,\lambda}(h) \mid h \in \mathcal{V}^c_\Omega\}$, of real numbers such that the Berezin measure (4.14) satisfies*

$$\langle \varphi_{h_1}\, \varphi_{h_2} \cdots \varphi_{h_{2N}} \rangle^{P_o}_{\Omega,x^*,\beta^*,\lambda} = \zeta^{P_o}_{\Omega,\lambda}(h_1) \cdots \zeta^{P_o}_{\Omega,\lambda}(h_{2N})\, \langle \varphi_{h_1}\, \varphi_{h_2} \cdots \varphi_{h_{2N}} \rangle^+_{\Omega,x_c} \\ + R^{P_o}_{\Omega,x^*,\beta^*,\lambda}(\xi), \tag{4.15}$$

*where $\langle\,\cdot\,\rangle^+_{\Omega,x_c}$ is the Berezin measure from Equation (3.9), and*

1. *$x^* = e^{-2\beta^* J}$;*
2. *$x_c = \sqrt{2}+1$ is the critical isotropic weight for the Ising model on the square lattice;*
3. *$\zeta^{P_o}_{\Omega,\lambda}$ are uniformly bounded and there exist constants $\theta > 0$ and $\zeta_{\mathrm{bulk},\lambda} \in \mathbb{R}$ such that*

$$\zeta^{P_o}_{\Omega,\lambda}(h) = \zeta_{\mathrm{bulk},\lambda} + O((\mathrm{dist}(h,\partial\Omega))^{-\theta}) \tag{4.16}$$

   *uniformly on $\Omega$ and $P_o$ for sufficiently large $\mathrm{dist}(h,\partial\Omega)$;*

4. *for two different domains $\Omega$ and $\Omega'$ and associated external boundary conditions $P_o$ and $P'_o$, we have*

$$\zeta^{P_o}_{\Omega,\lambda}(h) - \zeta^{P'_o}_{\Omega',\lambda}(h) = O((\mathrm{dist}(f,\Omega\triangle\Omega'))^{-\theta}) + O((\mathrm{dist}(f,P_o\triangle P'_o))^{-\theta}); \tag{4.17}$$

5. *and $R^{P_o}_{\Omega,x^*,\beta^*,\lambda}(\xi) = O(\min_{1\le i\ne j\le 2N}|h_i - h_j|^{-1-\theta})$ uniformly on $\Omega$ and $P_o$.*

Applying Conjecture 4.5 to the process in Theorem 4.4, we see that

$$\frac{\langle \varphi_{\vec{\gamma}_n}\, \varphi_{v_h} \rangle^{\check{P}_o(n)}_{\Omega\setminus\tilde{\gamma}_{[0,n]},x^*,\beta^*,\lambda}}{\langle \varphi_{\vec{\gamma}_n)}\, \varphi_{v_{h_{\mathrm{out}}}} \rangle^{\check{P}_o(n)}_{\Omega\setminus\tilde{\gamma}_{[0,n]},x^*,\beta^*,\lambda}}$$

$$= \frac{\zeta^{\check{P}_o(n)}_{\Omega\setminus\tilde{\gamma}_{[0,n]},\lambda}(\vec{\gamma}_n)\, \zeta^{\check{P}_o(n)}_{\Omega\setminus\tilde{\gamma}_{[0,n]},\lambda}(h)\, \langle \varphi_{\vec{\gamma}_n}\, \varphi_h \rangle^+_{\Omega,x_c} + R^{\check{P}_o(n)}_{\Omega\setminus\tilde{\gamma}_{[0,n]},x^*,\beta^*,\lambda}(\vec{\gamma}_n, h)}{\zeta^{\check{P}_o(n)}_{\Omega\setminus\tilde{\gamma}_{[0,n]},\lambda}(\vec{\gamma}_n)\, \zeta^{\check{P}_o(n)}_{\Omega\setminus\tilde{\gamma}_{[0,n]},\lambda}(h_{\mathrm{out}})\, \langle \varphi_{\vec{\gamma}_n}\, \varphi_{h_{\mathrm{out}}} \rangle^+_{\Omega,x_c} + R^{\check{P}_o(n)}_{\Omega\setminus\tilde{\gamma}_{[0,n]},x^*,\beta^*,\lambda}(\vec{\gamma}_n, h_{\mathrm{out}})}$$

$$= \frac{\zeta^{\check{P}_o(n)}_{\Omega \setminus \tilde{\gamma}_{[0,n]}, \lambda}(h)}{\zeta^{\check{P}_o(n)}_{\Omega \setminus \tilde{\gamma}_{[0,n]}, \lambda}(h_{\text{out}})} \frac{\langle \varphi_{v(\vec{\gamma}_n, \gamma_n)} \varphi_h \rangle^+_{\Omega, x_c}}{\langle \varphi_{v(\vec{\gamma}_n, \gamma_n)} \varphi_{h_{\text{out}}} \rangle^+_{\Omega, x_c}} + O([\text{dist}(\vec{\gamma}_n, h)]^{-\theta}) + O([\text{dist}(\vec{\gamma}_n, h_{\text{out}})]^{-\theta})$$

$$= \frac{\zeta_{\text{bulk}, \lambda}}{\zeta^{P_o}_{\Omega, \lambda}(h_{\text{out}})} \frac{\langle \varphi_{v(\vec{\gamma}_n, \gamma_n)} \varphi_h \rangle^+_{\Omega, x_c}}{\langle \varphi_{v(\vec{\gamma}_n, \gamma_n)} \varphi_{h_{\text{out}}} \rangle^+_{\Omega, x_c}} + O([\text{dist}(\tilde{\gamma}_{[0,n]} \cup \partial\Omega, h)]^{-\theta}) + O([\text{dist}(\tilde{\gamma}_{[0,n]}, h_{\text{out}})]^{-\theta})$$

in a locally uniform fashion, so that Conjecture 4.5 is sufficient to imply that the martingale observable in Theorem 4.4 has the same scaling limit as the corresponding one for the planar Ising model, up to an innocuous prefactor.

**Remark 4.6.** Equation (4.15) has the same form as existing results about correlation functions of the models under consideration [GGM12, AGG23, CGG24], except that here the prefactors are not constant. In the special case where all of $h_1, \ldots, h_{2N}$ lie on the boundary of the half-plane, Conjecture 4.5 should follow from a straightforward generalization of the proof in [CGG24][8].

In a constructive renormalization group treatment, the prefactors $\zeta^{P_o}_{\Omega, \lambda}(h)$ can be given as sums over collections of geometrical objects which contact $h$, which give a contribution decaying in their diameter and depend on the (difference in) boundary conditions only if they cross the (relevant portion of the) boundary, giving rise to Equations (4.16) and (4.17). This identification of the terms which are affected by the boundary conditions is also along similar lines to that of [Gre23] — although the geometrical objects considered there are trajectories of random walks or Brownian motions rather than the more varied collections of objects which would be involved here.

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
