# Peer review of "On the spin interface distribution for non-integrable variants of the two-dimensional Ising model"

_SciPost Physics_

## Round 3 · Referee Report · Anonymous (Referee 1) · 2025-1-6

Report

The paper by Greenblatt and Peltola concerns the critical behavior of a class of 2D Ising models obtained by perturbing the standard nearest neighbor model by a small finite range interaction of strength $\lambda$. In this setting, the first author and collaborators, in a series of previous works based on rigorous Renormalization Group (RG) methods, proved that, for $\lambda$ sufficiently small, the bulk energy correlation functions and the boundary spin correlations converge to the same scaling limit as the nearest neighbor model at its critical point, up to a finite multiplicative factor. In this paper, the goal is to understand the scaling limit of the interface induced by Dobrushin boundary conditions (DBC) associated with two marked points on the boundary of the domain, separating the region of the boundary corresponding to + boundary conditions from that with - boundary conditions. In the nearest neighbor case, $\lambda=0$, the problem has been solved and the interface distribution proved to converge to the (conformally covariant) Schramm-Loewner evolution process SLE(3). A key ingredient in the proof is the definition of an observable depending on the domain shape and a point in the interior; the definition involves the exploration path from one of the marked points on the boundary associated with the DBCs to the marked point in the interior, and in particular its winding. Crucially, this observable satisfies a martingale property, which allows one to control the Loewner driving function of the interface, and eventually, in combination with a proof of the existence of the scaling limit of the martingale observable, allows one to prove that the scaling limit of the interface is a specific SLE process. To date, essentially all these steps are proved in a way too heavily relying on the integrability properties of the nearest neighbor model. The present paper has the merit to outline a possible proof of convergence of the interface to SLE in the non-integrable setting, by defining a martingale observable for the perturbed model and by showing that such an observable admits a local representation in terms of the underlying fermionic fields that can be used to represent the Gibbs measure of the system. More preciely, it is the ratio of two two-point fermionic correlation functions in suitable (irregular) domains, similar to those studied earlier via RG methods by the first author and collaborators. An important difference is that in earlier works such correlation functions were studied in simple and regular domains, such as the full plane, a rectangular torus or a rectangular cylinder; the generalization via RG techniques to irregular domains such as those required here is not obvious. Still, on the basis of such previous results, the authors propose at a conjectural level a form for the scaling of the two point functions involved in the martingale observable, with a specific control on the error term at finite lattice spacing. This would be sufficient for proving convergence of the interface distrubution to the same scaling limit as for the nearest neighbor model.

I read the paper under review with interest: not only it proposes a concrete strategy for attacking an important open problem such as the convergence to SLE of the critical Ising interfaces separating + and - phases beyond the integrable case, but it does so by connecting two complementary approaches used so far for studying the problem, a probabilistic one (involving also techniques of discrete holomorphicity) due to Chelkak, Hongler, Smirnov, Izyurov and others, and one based on controlling perturbations of a Gaussian "fermionic Gibbs measure" via RG techniques due to Giuliani, Greenblatt, Mastropietro and others. The connection itself has interest and merit and may be useful to establish further connections among the two approaches.

This said, the paper also suffers from a number of limitations: first, and most importantly, the original results of the paper are quite limited. In my understanding, they consist in defining the appropriate observable in the non-integrable case and to show that it satisfies the martingale property; and to show that this observable admits a local representation in terms of fermionic fields. This is interesting, but maybe not at the level of a high profile research journal; rather, I tend to think that the paper is more appropriate for a review-style journal. However, I must admit that I don't know well the level of SciPost Physics; reading the description, its publications "provide details on groundbreaking results" that "have to meet our highest editorial standards", and I think that these requirements don't match with the results of the paper under review. On the other hand, it must be said that the two required expectations indicated by the authors, namely, to: "Provide a novel and synergetic link between different research areas", and "Open a new pathway in an existing or a new research direction, with clear potential for multi-pronged follow-up work" are fully met by the paper (as discussed in the previous paragraph). Therefore, the editorial board may consider to publish the paper even without any new "groundbreaking result" presented. I leave this choice to the editors.

There are other problems, which are less relevant, because they could be corrected in a revision, that also induce me not to recommend the paper for SciPost Physics in the present version, but only, possibly, after major revisions: - one is that the language is in my view too probabilistic: the authors assume the reader to be acquainted with the notions of martingale, filtration, pre-compactness, to mention a few, which I don't think are standard in theoretical physics. Their definition and usefulness (in particular that of martingale and filtration) should be briefly recalled - more importantly, some technical aspects of the discussion suffer for lack of details or imprecisions. Two important aspects that in my view are not well presented are the following: 1. The definition of the martingale observables in the $\lambda=0$ and $\lambda\neq0$ cases are not completely clear to me: more precisely, what I find unclear is why they are martingales with respect to the relevant Gibbs measures (the one of the nearest neighbor ising model, and the one of the perturbed model with $\lambda\neq0$, respectively); moreover, I don't understand how much freedom one has in definining the observable. In fact, the definition given by the authors appears to be valid for any family of functions $W^\xi$ parametrized by $\xi$ from the space of contours $\mathcal P^\xi_\Omega$ to the complex numbers, provided the phase of $W^\xi$ is constant over $\mathcal P^\xi_\Omega$ for any given $\xi$. If I understand correctly, in such a general setting, the process $M_W(h)$ defined in eq.(2.11) is a martingale with respect to the probability measures in eq.(2.10). However, I imagine that, in order for $M_W$ to be useful in a proof of convergence of the interfaces to an SLE process, the distribution associated with the probability measures in (2.10) should be the same as the one associated with the Gibbs distribution with the correct Ising Hamiltonian. At $\lambda=0$, this is briefly commented after eq.(2.10). In the general $\lambda\neq0$ I don't see any proof or discussion. Is it obvious? Isn't this the main result of the paper? Moreover, are there other possible choices for $W^\xi$ such that $M_W$ is a martingale with respect to the correct distribution? What are more precisely the constraints that $W^\xi$ has to satisfy in order for this to be the case? Do these constraints specify $W^\xi$ essentially uniquely or not? 2. It is unclear to me why, even heuristically, Conjecture 4.5 should be valid. The conjecture has a very precise form, which involves the prefactors in eq.(4.15) with a specific behavior as functions of $h,\Omega$ and $P_o$ (as described in eqs.(4.16)-(4.17)) and a remainder, with the specific behavior described in item 5 of the conjecture. Such precise and detailed properties seem to indicate that the authors have in mind a concrete strategy to prove them, modulo some technical issues that at the moment cannot solve. It would be useful to explain this to the reader, who otherwise is left with the impression that the conjecture is proposed a bit randomly. The only comments in this sense are contained in Remark 4.6, which however limits itself to say that in the case that the domain is the half-plane, the proof should be a straightforward generalization of the one in [CGG24]. However, the relevant domains the authors are interested in are very far from being as regular as the half-plane. Is there any concrete hope of proving the same in generic domains, such as those generated by a growing interface from a given nice domain $\Omega$? Is the representation vaguely mentioned as "sums over collections of geometrical objects which contact $h$" expected to be true also in the general case? Can the representation be described more precisely? At the moment I don't find it really comprehensible.

I have several other minor comments that are listed below.

  • p.2: as mentioned above, in the introduction the authors seem to assume that the reader is acquainted with the notions of martingale, filtration, martingale property (see l.7-8), of domain Markov property (see l.-10) and of pre-compactness (see l.-7), which are all non-standard outside the probability community and should be clarified.
  • p.2, l.-5, "... (FKG) inequalities: are these available for the model considered by the authors at least for a given sign of $\lambda$ and a secific class of $V(X)$? (I think so, and it should probably be mentioned). Would FKG be useful in a putative proof of convergence to SLE or not? That is, would Conjecture 4.5 be sufficient ``for everything" or would FKG be helpful (or needed) anyway?
  • p.3, eq.(1.1): at a first reading the use of the grey dots indexing the symbols $v,w,\mathcal E$ looks a bit misterious, I would at least mention that the apparently unnatural notation will make better sense below, after having introduced the various sites of the original lattice, its dual, etc
  • p.4, l.-3, "hull of the segment": what is it? And what does it mean to be measurable or not measurable with respect to the hull of the segment?
  • p.4, footnote 3 sounds a bit misterious: why is it the case that interacting dimer models are different? In which sense? And why to mention this class of models here? There are plenty of systems that have qualitatively different scaling limits than Ising, still expected to be conformally invariant.
  • p.5, third paragraph, "For the square lattice it is possible to guarantee ... In many other cases, the simplest scenario ..." and footnote 4. I don't understand why in a situation such as that considered by the authors, where the couplings are translationally invariant, the effective couplings should be inhomogeneous and even resembling a disordered case such as that mentioned in footnote 4. Depending on the choice of the lattice, there could be some local inhomogeneity, but nothing very serious; I would expect that after some coarse graining the inhomogeneities should effectively disappear without affecting the large distance behavior of the system.
  • p.8, l.16, "as long as ONE spin on the boundary...". Why just one? Doesn't one need to fix all the spins on the boundary in order to specify $\sigma$ given the contour configuration?
  • p.8, eq.(2.2) and preceding line: I think that there is a mistake in the indices: in eq.(2.2) the label of $x$ is $e$, while in the preceding line it is $e$ with an upper dot.
  • p.8, l.-2. I think that the symbol $2^{\mathcal E_\Omega}$ has not been defined yet (it is defined below, at p.11, l.-2)
  • p.9, two lines before (2.5): I'm a bit confused about the possible constraints that the choice of $(h_{in}, h_{out})$ should satisfy in order for the various definitions to make sense. Here the only condition mentioned is $h_{in}\neq h_{out}$. Shouldn't one also require something about $e(h_{in})$ and $e(h_{out})$ (e.g., that they should be - or not be - in $\mathcal E_{\Omega}$)? Moreover, can $f(h_{in})=f(h_{out})$?
  • p.9, caption of Fig.2.2, "While the marked dual edges...": I don't see any marked edge. Actually, I would find very useful to indicate in the figure what are the edges $e(h_{in})$ and $e(h_{out})$, thus clarifying, in particular, whether they belong or not to $\mathcal E_{\Omega}$.
  • p.9, l.1 after the caption: $\xi=(f(h_{in}),f(h_{out}))$ should be $\xi=(h_{in}, h_{out})$. Moreover, the notion of admissible boundary conditions is not very clear, some examples would be useful. E.g., it would be nice to add a possible choice of $\Omega_o$ in Fig.2.2 and exhibit (possibly in another figure) a choice of non-admissible $h$, clarifying why it is not so.
  • p.11, Definition 2.2: even though the sequence $\gamma_n$ is labelled by $n\in \mathbb Z_{\ge 0}$, it doesn't look to me to be semi-infinite (due to the stopping time in item 1 of the definition). Moreover, in item 3, I find the notation ${\gamma_n,\gamma_{n+1}}={f(\vec\gamma_n)\pm1/2, f(\vec\gamma_n)\pm i/2}$ unclear, a figure would be better. By the way, why so the authors refer to their choice as to the NE/SW rule? I would have rather called it NW/SE. Look at 2.5 and move along the interface following the numbers indicated there: the crossings where the rule must be applied produce the following corners: 1-2 (which corresponds to first moving east then south, so that I would call it ES), 4-5 (which corresponds to first moving north then west, so that I would call it NW), 5-6 (which corresponds to first moving west then north, so that I would call it WN), and 8-9 (which corresponds to first moving south then east, so that I would call it SE).
  • p.13, eqs.(2.10) and (2.12): are the two probability distributions related, or the same? Please clarify (possibly before or after eq.(2.12)). As mentioned in my main remarks above, it would be useful to better explain when and why these distributions coincide with those induced by the Gibbs measure, and what are the constraints induced on $W^\xi$ by the requirement that the distributions coincide.
  • p.13, l.4-6 after eq.(2.10): "gives rise to a martingale": recall what a martingale is. Moreover, the sentence "conditional expectations of a random variable given an increasing sequence of sigma-algebras form a tautological martingale" sounds mysterious, I don't understand what it means.
  • p.13, l.1 after eq.(2.11). Similarly to previous comment: the sentence "is a local martingale with respect to the natural filtration generated by $\gamma$" is not comprehensible without defining what is a "local martingale" and the "natural filtration"
  • p.14, last paragraph of Sect.2: I find also this discussion confusing. It seems to suggest that even under the constraint that the probability distribution induced by eq.(2.10) coincides with the appropriate Gibbs measure, there are many possible choices of $W^\xi$. Once again: can the authors specify more concretely what are the possible choices? If there are many, why do the authors in section 4 propose (without comments) just one specific choice of $W^\xi$?
  • p.16, Lemma 3.1: please specify that the symbol $1_{\mathcal P_{\Omega}}(X)$ indicates the characteristic function of the event $X\in \mathcal P_{\Omega}$
  • p.18, four lines before Lemma 3.6, "is a (local) martingale with repect to the exploration process associated with the Gibbs measure ..." Once again: why? What does it prevent to use the same $F_W$ for the model with $\lambda\neq0$?
  • p.20, l.1-5 and Remark 3.8: so what is the conclusion of these remarks? Is the proof of convergence complete with the current definition of $M_W$ or not? Is Remark 3.8 a proof or not?
  • p.20, eq.(3.21): how does the two-point function $<\varphi_{\vec\gamma_n}\varphi_{h_3}>^+{\Omega\setminus\vec\gamma$},\underline x behave at large distances? (This connects with my last comment below)
  • p.22, one line before (4.6), "gives rise to the martingale observable": I presume that the point is that the observable is a martingale with respect to the interacting Gibbs measure (otherwise, why to define $W$ as in eq.(4.4)): this should be specified, and possible proved (or, if the proof is obvious, this should be stated - and clarified why it is obvious).
  • p.26, statement of Conjecture 4.5: in addition to the comments made above before the list of minor comments , a couple more comments about item 3: what does it mean "uniformly bounded"? Does the notion just refer to an upper bound on the absolute value, or also to the fact that these numbers cannot be too close to zero? Moreover, I guess that $\zeta_{bulk,\lambda}$ (and possibly also the position-dependent factors) cannot be zero, this should be specified
  • p.26, last displayed equation: I think that in the last line, the domain $\Omega$ labelling the kets in the symbol for the two-point functions (both the one at the numerator and the one at the denominator) should be $\Omega\setminus\tilde\gamma_{[0,n]}$
  • p.27, displayed equation: I don't understand how the stated size of the error terms follows from the conjecture: in order to control the error terms one needs a control on the ratio between the function $R^{...}_{...}(\vec \gamma_n,h)$ and the two-point function $<\varphi_{\vec\gamma_n},h>^+_{...}$. I cannot find anything stated about the behavior of the two-point function, which is, by the way, needed for general $h$ and general shape of the interface. Can the authors please give details about this point?

Recommendation

Ask for major revision

---

## Round 3 · Referee Report · Anonymous (Referee 2) · 2025-4-16

Strengths

1. The paper is overall very well written, thorough, and self-contained. The introduction, in particular, is of excellent quality.
2. It identifies a martingale observable for the non-nearest-neighbour Ising model, representing a potential step toward proving conformal invariance for non-integrable Ising models — at least for those not too far from the critical non-nearest-neighbour case. This is achieved by identifying the Grassmann representation of the standard Ising martingale observable on the critical nearest-neighbour square lattice, and by leveraging the fact that the Grassmann representation naturally extends beyond the nearest-neighbour setting.
3. The authors formulate precise conjectures about the scaling limit, supported by explicit formulae. These are consistent with previous work by the first author, who computed correlations for the non-integrable Ising model in simple, explicit domains.

Weaknesses

1. The principal weakness of the paper lies in the absence of an explicit convergence statement. In models such as loop-erased random walk, percolation, dimers, the Ising model, or the harmonic explorer, identifying a martingale observable is a crucial but relatively small part of the overall effort required to establish conformal invariance—typically no more than 10% of the work. While the observable identified here represents a very interesting first step, it is worth noting that martingale observables are already known for all FK nearest-neighbour models, yet have not led to any significant conformal invariance results in those cases.
2. Convergence results for correlations in non-integrable models are currently available only in very specific and regular domains—and even then, not all of them. In contrast, tackling the interface convergence to SLE would require a much deeper understanding of scaling limits in arbitrary, potentially rough domains, as those are precisely the types of domains generated by the (fractal) exploration process in the limit. At present, I do not see how the proposed approach could succeed without substantial additional work on the convergence side. Such machinery appears to be both necessary and currently missing (as noted by the authors in their concluding remarks).

Report

The present paper, in my view, constitutes a very interesting note that opens up valuable discussions and perspectives regarding the challenge of proving conformal invariance in non-integrable Ising models. It is generally well written and provides a clear, concise presentation of the Grassmann approach for the nearest-neighbour model, with a particular emphasis on the treatment of the martingale observable. That said, as the authors themselves acknowledge, the path toward establishing full conformal invariance remains long and will require substantial further input. For this reason, I do not believe the paper, in its current form, is ready for publication. It may be more productive to let the broader research program mature further, potentially leading to a more comprehensive and impactful contribution in the future.

Recommendation

Ask for major revision

---

## Editorial Decision

awaiting_resubmission